# Neural Pfaffians: Solving Many Many-Electron Schrödinger Equations

**Nicholas Gao, Stephan Günnemann**
{n.gao,s.guennemann}@tum.de
Department of Computer Science & Munich Data Science Institute
Technical University of Munich

## Abstract

Neural wave functions accomplished unprecedented accuracies in approximating the ground state of many-electron systems, though at a high computational cost. Recent works proposed amortizing the cost by learning generalized wave functions across different structures and compounds instead of solving each problem independently. Enforcing the permutation antisymmetry of electrons in such generalized neural wave functions remained challenging as existing methods require discrete orbital selection via non-learnable hand-crafted algorithms. This work tackles the problem by defining overparametrized, fully learnable neural wave functions suitable for generalization across molecules. We achieve this by relying on Pfaffians rather than Slater determinants. The Pfaffian allows us to enforce the antisymmetry on arbitrary electronic systems without any constraint on electronic spin configurations or molecular structure. Our empirical evaluation finds that a single neural Pfaffian calculates the ground state and ionization energies with chemical accuracy across various systems. On the TinyMol dataset, we outperform the 'gold-standard' CCSD(T) CBS reference energies by $1.9\,\mathrm{m}E_\mathrm{h}$ and reduce energy errors compared to previous generalized neural wave functions by up to an order of magnitude.

## 1 Introduction

Solving the electronic Schrödinger equation is at the heart of computational chemistry and drug discovery. Its solution provides a molecule's or material's electronic structure and energy (Zhang et al., 2023). While the exact solution is infeasible, neural networks have recently shown unprecedentedly accurate approximations (Hermann et al., 2023). These neural networks approximate the system's ground-state wave function $\Psi : \mathbb{R}^{N_\mathrm{e} \times 3} \to \mathbb{R}$, the lowest energy state, by minimizing the energy $\langle \Psi | \hat{H} | \Psi \rangle$, where $\hat{H}$ is the Hamiltonian operator, a mathematical description of the system. While such neural wave functions are highly accurate, training has proven computationally intensive.

Gao & Günnemann (2022) have shown that training a generalized neural wave function on a large class of systems amortizes the cost. However, their approach is limited to different geometric arrangements of the same molecule. Subsequent works eliminated this limitation by introducing hand-crafted algorithms (Gao & Günnemann, 2023a) or heavily relying on classical Hartree-Fock calculations (Scherbela et al., 2023). Both impose strict, non-learnable mathematical constraints and prior assumptions that may not always hold, limiting their generalization and accuracies. Hand-crafted algorithms only work for a limited set of molecules, in particular organic molecules near equilibrium, while the reliance on Hartree-Fock empirically results in degraded accuracies.

In this work, we propose the Neural Pfaffian (NeurPf) to overcome these limitations. As suggested by its name, NeurPf uses Pfaffians to define a superset of the previously used Slater determinants to enforce the fermionic antisymmetry. The Pfaffian lifts the constraint on the number of molecular orbitals from Slater determinants (Szabo & Ostlund, 2012), enabling overparametrized wave functions

with simpler and more accurate generalization. Compared to Globe (Gao & Günnemann, 2023a), the absence of hand-crafted algorithms enables the modeling of non-equilibrium, ionized, or excited systems. By being fully learnable without fixed Hartree-Fock calculations like TAO (Scherbela et al., 2024), NeurPf achieves significantly lower variational energies. Our empirical results show that NeurPf can learn all second-row elements' ground-state, ionization, and electron affinity potentials with a single wave function. Further, we demonstrate that NeurPf's accuracy surpasses Globe on the challenging nitrogen dimer with seven times fewer parameters while not suffering from performance degradations when adding structures to the training set. On the TinyMol dataset, NeurPf surpasses the highly accurate reference CCSD(T) CBS energies on the small structures by $1.9\,\mathrm{m}E_\mathrm{h}$ and reduces errors compared to TAO by factors of 10 and 6 on the small and large structures, respectively.

## 2 Quantum chemistry

Quantum chemistry aims to solve the time-independent Schrödinger equation (Foulkes et al., 2001)

$$\hat{H}\left|\Psi\right\rangle = E\left|\Psi\right\rangle \tag{1}$$

where $\Psi : \mathbb{R}^{N_\uparrow \times 3} \times \mathbb{R}^{N_\downarrow \times 3} \to \mathbb{R}$ is the electronic wave function for $N_\uparrow$ spin-up and $N_\downarrow$ spin-down electrons, $\hat{H}$ is the Hamiltonian operator, and $E$ is the system's energy. To ease notation, if not necessary, we omit spins in $\Psi$ and treat it as $\Psi : \mathbb{R}^{N_\mathrm{e} \times 3} \to \mathbb{R}$ where $N_\mathrm{e} = N_\uparrow + N_\downarrow$. The Hamiltonian $\hat{H}$ for molecular systems, which we are concerned with in this work, is given by

$$\hat{H} = -\frac{1}{2}\sum_{i=1}^{N_\mathrm{e}}\sum_{k=1}^{3}\frac{\partial^2}{\partial \vec{r}_{ik}^2} + \sum_{j>i}^{N_\mathrm{e}}\frac{1}{\|\vec{r}_i - \vec{r}_j\|} - \sum_{i=1}^{N_\mathrm{e}}\sum_{m=1}^{N_\mathrm{n}}\frac{Z_m}{\|\vec{r}_i - \vec{R}_m\|} + \sum_{n>m}^{N_\mathrm{n}}\frac{Z_m Z_n}{\|\vec{R}_m - \vec{R}_n\|} \tag{2}$$

with $\vec{r}_i \in \mathbb{R}^3$ being the $i$th electron's position, and $\vec{R}_m \in \mathbb{R}^3, Z_m \in \mathbb{N}_+$ being the $m$th nucleus' position and charge. The wave function $\Psi$ describes the behavior of electrons in the system defined by the Hamiltonian $\hat{H}$. As the square of the wave function $\Psi^2$ is proportional to the probability density $p(\vec{r}) \propto \Psi^2(\vec{r})$ of finding the electrons at positions $\vec{r} \in \mathbb{R}^{N_\mathrm{e} \times 3}$, its integral must be finite:

$$\int \Psi(\vec{r})^2 \mathrm{d}\vec{r} < \infty. \tag{3}$$

Further, as electrons are indistinguishable half-spin fermionic particles, the wave function must be antisymmetric under any same-spin electron permutation $\tau$:

$$\Psi\left(\tau^\uparrow(\vec{r}^\uparrow), \tau^\downarrow(\vec{r}^\downarrow)\right) = \mathrm{sgn}(\tau^\uparrow)\mathrm{sgn}(\tau^\downarrow)\Psi(\vec{r}). \tag{4}$$

To enforce this constraint, the wave function is typically defined as a so-called Slater determinant of $N_\uparrow + N_\downarrow$ integrable so-called orbital functions $\phi_i : \mathbb{R}^3 \to \mathbb{R}$:

$$\Psi_{\mathrm{Slater}}(\vec{r}) = \det\left[\phi_j^\uparrow(\vec{r}_i^\uparrow)\right]\det\left[\phi_j^\uparrow(\vec{r}_i^\downarrow)\right] = \det \Phi^\uparrow(\vec{r}^\uparrow)\det \Phi^\downarrow(\vec{r}^\downarrow). \tag{5}$$

Note that for the determinant to exist, one needs exactly $N_\uparrow$ up and $N_\downarrow$ down orbitals $\phi_j^\uparrow$ and $\phi_j^\downarrow$.

In linear algebra, Eq. (1) is an eigenvalue problem, where we look for the eigenfunction $\Psi_0$ with the lowest eigenvalue $E_0$. In Variational Monte Carlo (VMC), this is solved by applying the variational principle, which states that the energy of any trial wave function $\Psi$ upper bounds $E_0$:

$$E_0 \leq \frac{\langle\Psi|\hat{H}|\Psi\rangle}{\langle\Psi^2\rangle} = \frac{\int \Psi(\vec{r})\hat{H}\Psi(\vec{r})\mathrm{d}\vec{r}}{\int \Psi^2(\vec{r})\mathrm{d}\vec{r}}. \tag{6}$$

By plugging in the probability distribution from Eq. (3), we can rewrite Eq. (6) as

$$E_0 \leq \mathbb{E}_{p(\vec{r})}\left[\Psi^{-1}(\vec{r})\hat{H}\Psi(\vec{r})\right] = \mathbb{E}_{p(\vec{r})}\left[E_L(\vec{r})\right], \tag{7}$$

with $E_L(\vec{r}) = \Psi(\vec{r})^{-1}\hat{H}\Psi(\vec{r})$ being the so-called local energy. The right-hand side of Eq. (7) is known as the variational energy. As Eq. (7) does not require $\Psi$ to be an analytic function, we can approximate the energy of any valid wave function $\Psi$ with samples drawn from $p(\vec{r})$. If we pick a

parametrized family of wave functions $\Psi_\theta$, we can optimize the parameters $\theta$ to minimize the VMC energy by following the gradient of the variational energy

$$\nabla_\theta = \mathbb{E}_{p(\vec{\mathbf{r}})} \left[ (E_L(\vec{\mathbf{r}}) - \mathbb{E}_{p(\vec{\mathbf{r}})} [E_L(\vec{\mathbf{r}})]) \nabla_\theta \log \Psi_\theta(\vec{\mathbf{r}}) \right], \tag{8}$$

where we approximate all expectations by Monte Carlo sampling (Ceperley et al., 1977).

**Neural wave functions** typically keep the functional form of Eq. (5) but replace the orbitals $\phi_i$ with learned many-electron orbitals $\phi_i^{\text{NN}} : \mathbb{R}^3 \times \mathbb{R}^{N_e \times 3} \to \mathbb{R}$ (Hermann et al., 2023). These many-electron orbitals $\phi_i^{\text{NN}}$ are implemented as different readouts of the same permutation-equivariant neural network. Multiplying each orbital by an envelope function $\chi_i : \mathbb{R}^3 \to \mathbb{R}$ that decays exponentially to zero at large distances enforces the finite integral requirement in Eq. (3).

**Generalized wave functions** solve the more general problem where the nucleus positions $\vec{\mathbf{R}}$ and charges $\mathbf{Z}$ are not fixed. Since the Hamiltonian $\hat{H}_{\vec{\mathbf{R}}, \mathbf{Z}}$ depends on the molecular structure $(\vec{\mathbf{R}}, \mathbf{Z})$, so does the corresponding ground state wave function $\Psi_{\vec{\mathbf{R}}, \mathbf{Z}}$. Note that we still work in the Born-Oppenheimer approximation, i.e., we treat the nuclei as classical point charges (Zhang et al., 2023). Given a dataset of molecular structures $\mathcal{D} = \{(\vec{\mathbf{R}}_1, \mathbf{Z}_1), ...\}$, the total energy $\sum_{(\vec{\mathbf{R}}, \mathbf{Z}) \in \mathcal{D}} \frac{\langle \Psi_{\vec{\mathbf{R}}, \mathbf{Z}} | \hat{H}_{\vec{\mathbf{R}}, \mathbf{Z}} | \Psi_{\vec{\mathbf{R}}, \mathbf{Z}} \rangle}{\langle \Psi_{\vec{\mathbf{R}}, \mathbf{Z}}^2 \rangle}$ is minimized to approximate the ground state for each structure. Typically, the dependence on $\vec{\mathbf{R}}, \mathbf{Z}$ is implemented by using a meta network that takes $\vec{\mathbf{R}}, \mathbf{Z}$ as inputs and outputs the parameters of the electronic wave function (Gao & Günnemann, 2022).

## 3 Related work

While attempts to enforce the fermionic antisymmetry in neural wave functions in less than $O(N_e^3)$ operations promise faster runtime than Slater determinants, the accuracy of these methods is limited (Han et al., 2019; Acevedo et al., 2020; Richter-Powell et al., 2023). Pfau et al. (2020) and Hermann et al. (2020) established Slater determinants for neural wave functions by demonstrating chemical accuracy on small molecules. Note, Eq. (5) may also be written via a block-diagonal matrix, i.e., $\Psi(\vec{\mathbf{r}}) = \det\left(\text{diag}(\Phi^\uparrow, \Phi^\downarrow)\right)$. Spencer et al. (2020)'s implementation further increased accuracies by parametrizing the off diagonals that were implicitly set to 0 before, with additional orbitals $\tilde{\Phi}$:

$$\Psi_{\text{Slater}}(\vec{\mathbf{r}}) = \det(\hat{\Phi}(\vec{\mathbf{r}})) = \det \begin{bmatrix} \Phi^\uparrow(\vec{\mathbf{r}}^\uparrow) & \tilde{\Phi}^\uparrow(\vec{\mathbf{r}}^\uparrow) \\ \tilde{\Phi}^\downarrow(\vec{\mathbf{r}}^\downarrow) & \Phi^\downarrow(\vec{\mathbf{r}}^\downarrow) \end{bmatrix}. \tag{9}$$

Several works confirmed the improved empirical accuracy of this approach (Gerard et al., 2022; Lin et al., 2021; Ren et al., 2023; Gao & Günnemann, 2023b, 2024). While later works refined the architecture to increase accuracy (von Glehn et al., 2023; Wilson et al., 2021, 2023), the use of Slater determinants mostly remained a constant, with two notable exceptions: Firstly, Lou et al. (2023) use AGP wave functions (Casula & Sorella, 2003; Casula et al., 2004) to formulate the wave function as $\Psi(\vec{\mathbf{r}}) = \det(\Phi^\uparrow) \det(\Phi^\downarrow) = \det(\Phi^\uparrow \Phi^{\downarrow T})$. This avoids picking exactly $N_\uparrow / N_\downarrow$ orbitals as $\Phi^\uparrow$ and $\Phi^\downarrow$ may be non-square but fails to generalize Eq. (9), we empirically verify the impact of this limitation in App. I. Secondly, Kim et al. (2023) introduced the combination of neural networks and Pfaffians, who demonstrated its performance on the ultra-cold Fermi gas. Though universal in theory, their parametrization yields no trivial adaption to molecular systems. In classical quantum chemistry, Bajdich et al. (2006, 2008) reported promising early results with Pfaffians in single-structure calculations for small molecules. In this work, we generalize Eq. (9) to Pfaffian wave functions that permit pretraining with Hartree-Fock calculations and generalization across molecules.

**Generalized wave functions.** Scherbela et al. (2022) started this research with a weight-sharing scheme between wave functions. These still had to be reoptimized for each structure. Later, Gao & Günnemann (2022, 2023b) proposed PESNet, a generalized wave function for energy surfaces allowing joint training without reoptimization. Subsequent works extended PESNet to different compounds where the main challenge is parametrizing exactly $N_\uparrow + N_\downarrow$ orbitals, such that the orbital matrix in Eq. (9) stays square. The problem of finding these orbitals was formulated into a discrete orbital selection problem. Gao & Günnemann (2023a)'s hand-crafted algorithm accomplishes this by selecting orbitals via a greedy nearest neighbor search. In contrast, Scherbela et al. (2024, 2023) use the lowest eigenvalues of the Fock matrix as selection criteria. Both introduce non-learnable constraints, limiting generalization or sacrificing accuracy. NeurPf avoids the selection problem by introducing an overparametrization when enforcing the exchange antisymmetry.

# 4 Neural Pfaffian

Previous generalized wave functions build on Slater wave functions and attempt to adjust the orbitals $\phi_i$ to the molecule. Slater determinants were chosen due to their previously demonstrated high accuracy. However, they require exactly $N_\uparrow + N_\downarrow$ orbitals. While the nuclei allow inferring the total number of electrons $N_e$ of any stable, singlet state system, the spin distribution into $N_\uparrow$ and $N_\downarrow$ orbitals per atom is not readily available. Previous works implement this via a discrete selection of orbitals via non-learnable prior assumptions and constraints on the wave function; see Sec. 3.

Here, we present the Neural Pfaffian (NeurPf), a superset of Slater wave functions that preserves accuracy while relaxing the orbital number constraint. By not enforcing an exact number of orbitals, NeurPf is overparametrized with $N_o \geq \max\{N_\uparrow, N_\downarrow\}$ orbitals, avoiding discrete selections and making it a natural choice for generalized wave functions. Importantly, NeurPf can be pretrained with Hartree-Fock, which accounts for $> 99\%$ of the total energy (Szabo & Ostlund, 2012). We introduce NeurPf in four steps: (1) We introduce the Pfaffian and use it to define a superset of Slater wave functions. (2) We present memory-efficient envelopes that additionally accelerate convergence. (3) We introduce a new pretraining scheme for matching Pfaffian and Slater wave functions. (4) We discuss combining our developments to build a generalized wave function.

## 4.1 Pfaffian wave function

The Pfaffian of a skew-symmetric $2n \times 2n$ matrix $A$, i.e., $A = -A^T$, is defined as

$$\text{Pf}(A) = \frac{1}{2^n n!} \sum_{\tau \in S_{2n}} \text{sgn}(\tau) \prod_{i=1}^{n} A_{\tau(2i-1),\tau(2i)} \tag{10}$$

where $S_{2n}$ is the symmetric group of $2n$ elements. One may consider it a square root of the determinant of $A$ since $\text{Pf}(A)^2 = \det(A)$. An important property of the Pfaffian is $\text{Pf}(BAB^T) = \det(B)\text{Pf}(A)$ for any invertible matrix $B$ and skew-symmetric matrix $A$. In the context of neural wave functions, this means that if $A$ is an along both dimensions permutation equivariant function of the electron positions $\vec{r}$, $A(\tau(\vec{r})) = P_\tau A(\vec{r})P_\tau^T$, the Pfaffian of $A$ is a valid wave function that fulfills the antisymmetry requirement from Eq. (4):

$$\Psi(\tau(\vec{r})) = \text{Pf}(A(\tau(\vec{r}))) = \text{Pf}(P_\tau A(\vec{r})P_\tau^T) = \det(P_\tau)\text{Pf}(A(\vec{r})) = \text{sign}(\tau)\Psi(\vec{r}). \tag{11}$$

To compute the Pfaffian without evaluating the $2n!$ terms in Eq. (10), we implement the Pfaffian via a tridiagonalization with the Householder transformation as in Wimmer (2012).

There are various ways to construct $A$ (Bajdich et al., 2006, 2008; Kim et al., 2023). Here, we introduce a superset of Slater wave functions, enabling high accuracy on molecular systems. If $A$ is a skew-symmetric matrix, so is $BAB^T$ for any arbitrary matrix $B$. Thus, we can construct $\Psi_{\text{Pfaffian}}$ as

$$\Psi_{\text{Pfaffian}}(\vec{r}) = \frac{1}{\text{Pf}(A_{\text{Pf}})} \text{Pf}\left(\hat{\Phi}_{\text{Pf}}(\vec{r}) A_{\text{Pf}} \hat{\Phi}_{\text{Pf}}(\vec{r})^T\right) \tag{12}$$

where $A_{\text{Pf}} \in \mathbb{R}^{N_o \times N_o}$ is a learnable skew-symmetric matrix and $\hat{\Phi}_{\text{Pf}} : \mathbb{R}^{N_e \times 3} \to \mathbb{R}^{N_e \times N_o}$ is a permutation equivariant function like in Eq. (9). This construction elevates the need for having exactly $N_\uparrow/N_\downarrow$ orbitals as in Slater determinants. We may now overparametrize the wave function with $N_o \geq \max\{N_\uparrow, N_\downarrow\}$ orbitals, allowing for a more flexible and simpler implementation without needing discrete orbital selection. By choosing $\hat{\Phi}_{\text{Pf}} = \hat{\Phi}$, it is straightforward to see that Eq. (12) is a superset of the Slater determinant wave function in Eq. (9). Note that, like in Eq. (9), we parametrize two sets of orbital functions $\Phi_{\text{Pf}}$ and $\tilde{\Phi}_{\text{Pf}}$ and change their order for spin-down electrons to not enforce the exchange antisymmetry between different-spin electrons. As the normalizer $\text{Pf}(A_{\text{Pf}})$ is constant, we drop it going forward. As it is common in quantum chemistry (Szabo & Ostlund, 2012; Hermann et al., 2020), we use linear combinations of wave functions to increase expressiveness:

$$\Psi_{\text{Pfaffian}}(\vec{r}) = \sum_{k=1}^{N_k} c_k \Psi_{\text{Pfaffian},k}(\vec{r}). \tag{13}$$

We visually compare the schematic of the Slater determinant and Pfaffian wave functions in Fig. 1. In App. A, we discuss how to handle odd numbers of electrons such that $\hat{\Phi}_{\text{Pf}} A_{\text{Pf}} \hat{\Phi}_{\text{Pf}}^T$ has even

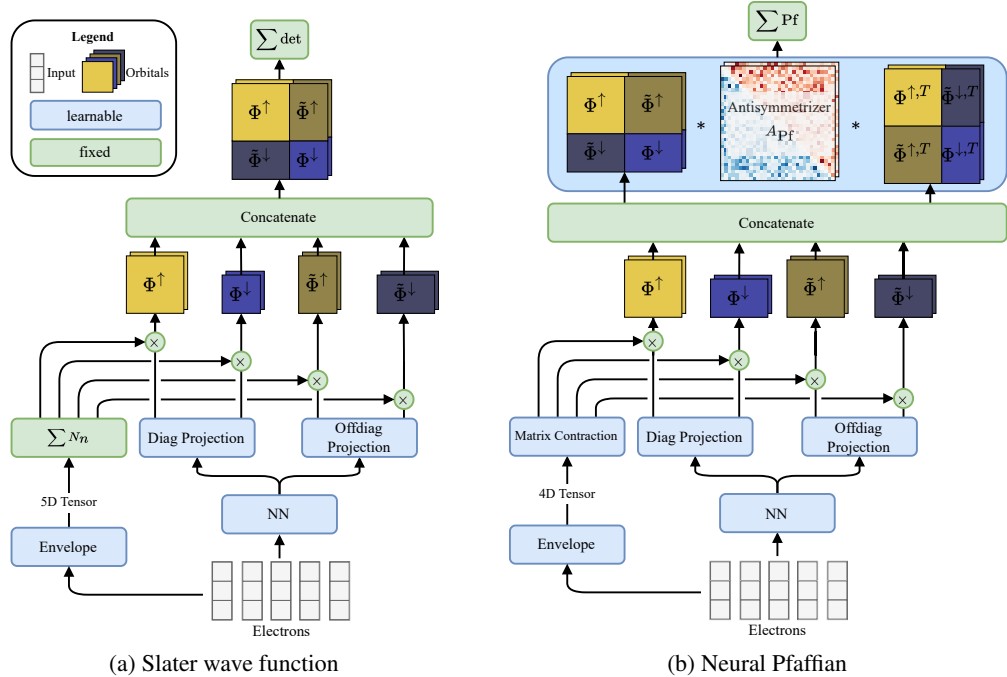

(a) Slater wave function        (b) Neural Pfaffian

Fig. 1: Schematic of the Slater determinant (1a) and our NeurPf (1b). Where the Slater formulation requires exactly $N_e$ orbital functions, the Pfaffian formulation works for any number $N_o \geq \max\{N_\uparrow, N_\downarrow\}$ of orbital functions, indicated by the rectangular orbital blocks.

dimensions. Like previous work (Pfau et al., 2020), we parametrize the orbital functions $\phi_i$ as a product of a permutation equivariant neural network $\boldsymbol{h} : \mathbb{R}^3 \times \mathbb{R}^{N \times 3} \to \mathbb{R}^{N_f}$ and an envelope function $\chi : \mathbb{R}^3 \to \mathbb{R}$:

$$\phi_{ki}(\vec{r}_j | \vec{\mathbf{r}}) = \chi_{ki}(\vec{r}_j) \cdot \boldsymbol{h}(\vec{r}_j | \vec{\mathbf{r}})^T \boldsymbol{w}_{ki} \cdot \eta_{ki}^{N_\uparrow - N_\downarrow} \tag{14}$$

with $\boldsymbol{w}_{ki} \in \mathbb{R}^{N_f}$ being a learnable weight vector, and $\eta_{ki}^{N_\uparrow - N_\downarrow} \in \mathbb{R}$ being a scalar depending on the spin state of the system, i.e., the difference between the number of up and down electrons. The envelope function $\chi$ ensures that the integral of the squared wave function is finite. For $\boldsymbol{h}$, we use Moon from Gao & Günnemann (2023a) thanks to its size consistency.

## 4.2 Memory-efficient envelopes

To satisfy the finite integral requirement on the square of $\Psi$ in Eq. (3), the orbitals $\phi$ are multiplied by an envelope function $\chi : \mathbb{R}^3 \to \mathbb{R}$ that exponentially decays to zero at large distances. We do not split spins here and work with $N_e = N_\uparrow + N_\downarrow$ to simplify the discussion, but, in practice, we would split the envelopes into two sets, one for $\Phi_{Pf}$ and one for $\tilde{\Phi}_{Pf}$. The envelope function is typically a sum of exponentials centered on the nuclei (Spencer et al., 2020). In Einstein's summation notation, the envelope function can be written as

$$\chi_{ki}(\vec{r}_{bj}) = \underbrace{\boldsymbol{\pi}_{kmi}}_{N_k \times N_n \times N_o} \cdot \underbrace{\exp(-\boldsymbol{\sigma}_{kmi} \|\vec{\mathbf{r}}_{bj} - \vec{\mathbf{R}}_m\|)_{bkmji}}_{N_b \times N_k \times N_n \times N_e \times N_o} \tag{15}$$

where $N_b$ denotes the batch size. Empirically, we found the tensor on the right side containing many redundant entries. Further, due to the nonlinearity of the exponential function, one cannot implement the envelope in a simple matrix contraction but has to materialize the full five-dimensional tensor. NeurPf amplifies this problem as $N_o \geq N_e$ whereas Slater determinants constraint $N_o = N_e$.

We use a single set of exponentials per nucleus instead of having one for each combination of orbital and nucleus. This reduces the number of envelopes per electron from $N_k \times N_n \times N_o$ to $N_k \times N_{env}$, where $N_{env} = N_n \times N_{env/nuc}$ is the number of envelope functions. In general, we pick $N_{env/nuc}$

such that $N_{\text{env}} \approx N_{\text{o}}$. These atomic envelopes are linearly recombined into molecular envelopes, effectively enlarging $\boldsymbol{\pi}$ to a $N_{\text{k}} \times N_{\text{o}} \times N_{\text{env}}$ tensor. Thanks to these rearrangements, we avoid constructing a five-dimensional tensor. Instead, we define the envelopes as

$$\chi_{ki}(\vec{r}_{bj}) = \underbrace{\boldsymbol{\pi}_{kni}}_{N_{\text{k}} \times N_{\text{env}} \times N_{\text{o}}} \cdot \underbrace{\exp(-\boldsymbol{\sigma}_{kn}\|\vec{r}_{bj} - \mathbf{R}_n\|)_{kbnj}}_{N_{\text{b}} \times N_{\text{k}} \times N_{\text{env}} \times N_{\text{e}}}. \tag{16}$$

Concurrently, Pfau et al. (2024) presented similar bottleneck envelopes. However, we found ours to converge faster and not yield numerical instabilities. We discuss this further in App. B and I.

### 4.3 Pretraining Pfaffian wave functions

Pretraining is essential in training neural wave functions and has frequently been observed to critically affect final energies (Gao & Günnemann, 2023a; von Glehn et al., 2023; Gerard et al., 2022). The pretraining aims to find orbital functions close to the ground state to stabilize the optimization. Traditionally, this is done by matching the orbitals of the neural wave function to the orbitals of a baseline wave function, typically a Hartree-Fock wave function $\Psi_{\text{HF}} = \det(\Phi_{\text{HF}})$, by solving

$$\min_{\boldsymbol{\theta}} \|\Phi_{\boldsymbol{\theta}} - \Phi_{\text{HF}}\|_2^2, \tag{17}$$

for the neural network parameters $\boldsymbol{\theta}$ (Pfau et al., 2020). Since our Pfaffian has $N_{\text{o}}$ orbitals while Hartree-Fock has $N_{\text{e}}$, we cannot directly apply this to our Pfaffian wave function. Further, as we predict orbitals per nucleus, our arbitrary orbital order may not align with Hartree-Fock.

We propose two alternative pretraining schemes for neural Pfaffian wave functions: one based on matching single-electron orbitals and one based on matching geminals, effectively two-electron orbitals. We need to expand the Hartree-Fock orbitals $\Phi^{\text{HF}}$ to $N_{\text{o}}$ orbitals to match the single-electron orbitals directly. We construct $\bar{\Phi}^{\text{HF}}$ by padding the extra $N_{\text{o}} - N_{\text{e}}$ orbitals with zeros. It can easily be verified that the wave function $\Psi_{\text{HF-Pf}} = \frac{1}{\text{Pf} \bar{A}_{\text{HF}}} \text{Pf}(\bar{\Phi}_{\text{HF}} \bar{A}_{\text{HF}} \bar{\Phi}_{\text{HF}}^T)$, is equivalent to the original Hartree-Fock wave function, i.e., $\Psi_{\text{HF-Pf}} = \Psi_{\text{HF}} = \det(\Phi_{\text{HF}})$ for any invertible skew-symmetric $A_{\text{HF}}$. Further, note that the multiplication of $\bar{\Phi}_{\text{HF}}$ with any matrix $T \in SO(N_{\text{o}})$ from the special orthogonal group does not change $\Psi_{\text{HF-Pf}}$. Thus, it suffices to match the single electron orbitals of $\hat{\Phi}_{\text{Pf}}$ and $\bar{\Phi}_{\text{HF}}$ up to a rotation $T \in SO(N_{\text{o}})$, yielding the following optimization problem:

$$\min_{\boldsymbol{\theta}} \min_{T \in SO(N_{\text{o}})} \|\hat{\Phi}_{\text{Pf}} - \bar{\Phi}_{\text{HF}} T\|_2^2. \tag{18}$$

We solve this alternatingly for $T$ and $\boldsymbol{\theta}$. To match the geminals $\hat{\Phi}_{\text{Pf}} A_{\text{Pf}} \hat{\Phi}_{\text{Pf}}^T$ and $\Phi_{\text{HF}} A_{\text{HF}} \Phi_{\text{HF}}^T$, we have to account for the fact that the choice of $A_{\text{HF}}$ is arbitrary as long as it is skew-symmetric and invertible. Again, we solve this optimization problem alternatingly by solving for $A_{\text{HF}} \in \mathbb{S} = \{A \in SO(N_{\text{e}}) : A = -A^T\}$ and $\boldsymbol{\theta}$:

$$\min_{\boldsymbol{\theta}} \min_{A_{\text{HF}} \in \mathbb{S}} \|\hat{\Phi}_{\text{Pf}} A_{\text{Pf}} \hat{\Phi}_{\text{Pf}}^T - \Phi_{\text{HF}} A_{\text{HF}} \Phi_{\text{HF}}^T\|_2^2. \tag{19}$$

While both formulations share the same minimizer, combining both yields the most stable results. We hypothesize that this is because the single-electron orbitals are more stable than the geminals and thus provide a better starting point for the optimization. In contrast, the latter provides a closer formulation of the neural network orbitals. Thus, we pretrain our neural Pfaffian wave functions by solving the optimization problem

$$\min_{\boldsymbol{\theta}} \left( \alpha \min_{T \in SO(N_{\text{o}})} \|\hat{\Phi}_{\text{Pf}} - \bar{\Phi}_{\text{HF}} T\|_2^2 + \beta \min_{A_{\text{HF}} \in \mathbb{S}} \|\hat{\Phi}_{\text{Pf}} A_{\text{Pf}} \hat{\Phi}_{\text{Pf}}^T - \Phi_{\text{HF}} A_{\text{HF}} \Phi_{\text{HF}}^T\|_2^2 \right) \tag{20}$$

with weights $\alpha, \beta \in [0, 1]$. To optimize over the special orthogonal group $SO(N_{\text{o}})$, we use the Cayley transform (Gallier, 2013). App. C further details the procedure.

### 4.4 Generalizing over systems

We now focus on generalizing the construction of our Pfaffian wave function for different systems. We accomplish the generalization similar to PESNet (Gao & Günnemann, 2022) by introducing a second neural network, the MetaGNN $\mathcal{M} : (\mathbb{R}^3 \times \mathbb{N}_+)^{N_n} \to \Theta$ that acts upon the molecular structure, i.e., nuclei positions and charges, and parametrizes the electronic wave function $\Psi_{\text{Pfaffian}} : \mathbb{R}^{N_e \times 3} \times \Theta \to \mathbb{R}$ for the system of interest. As architecture for the wave function and MetaGNN, we use the same architecture as in Gao et al. (2023a) with the exception being that we replace the Slater determinant with the Pfaffian as described in Sec. 4 and minor tweaks highlighted in App. D.4.

**Pfaffian.** To represent wave functions of different systems within a single NeurPf, we need to adapt the orbitals $\hat{\Phi}_{\mathrm{Pf}}$ and antisymmetrizer $A_{\mathrm{Pf}}$ from Eq. (12) to the molecule. In doing so, we must ensure $N_{\mathrm{o}} \geq \max\{N_{\uparrow}, N_{\downarrow}\}$. Otherwise, $\hat{\Phi}_{\mathrm{Pf}} A_{\mathrm{Pf}} \hat{\Phi}_{\mathrm{Pf}}^T$ is singular, and the wave function is zero. One may solve this by picking $N_{\mathrm{o}}$ large enough that $N_{\mathrm{o}} \geq \max\{N_{\uparrow}, N_{\downarrow}\}$ for all molecules in the dataset. However, this is computationally expensive, does not reuse known orbitals in the problem, and simply moves the problem to even larger systems. Instead, we grow the number of orbitals $N_{\mathrm{o}}$ with the system size by defining $N_{\mathrm{orb/nuc}}$ orbitals per nucleus, as depicted in Fig. 2. This allows us to transfer orbitals from smaller systems to larger systems. We only need to ensure that $N_{\mathrm{orb/nuc}}$ is larger than half the maximum number of electrons in a period, e.g., for the first period $N_{\mathrm{orb/nuc}} \geq 1$, for the second period $N_{\mathrm{orb/nuc}} \geq 5$.

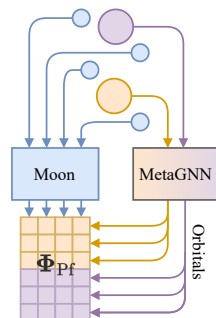

Fig. 2: Orbital parametrization per nucleus. ◯, ◯/◯ indicate electrons and nuclei, respectively.

The projection $\boldsymbol{W}$ from Eq. (14) and the envelope decays $\boldsymbol{\sigma}$ are parametrized by node embeddings, while the envelope weights $\boldsymbol{\pi}$ and the antisymmetrizer $A_{\mathrm{Pf}}$ are derived from edge embeddings. We predict a $N_{\mathrm{orb/nuc}} \times N_{\mathrm{f}}$ matrix per nucleus for $\boldsymbol{W}$ and a $N_{\mathrm{env/nuc}}$ vector per nucleus for $\boldsymbol{\sigma}$. For the edge parameters $\boldsymbol{\pi}$ and $A_{\mathrm{Pf}}$, we predict a $N_{\mathrm{env/nuc}} \times N_{\mathrm{orb/nuc}}$ and a $N_{\mathrm{orb/nuc}} \times N_{\mathrm{orb/nuc}}$ matrix per edge, respectively. These are concatenated into the $N_{\mathrm{env}} \times N_{\mathrm{o}}$ and $N_{\mathrm{o}} \times N_{\mathrm{o}}$ matrices $\boldsymbol{\pi}$ and $\hat{A}_{\mathrm{Pf}}$. The latter is antisymmetrized to get $A_{\mathrm{Pf}} = \frac{1}{2}(\hat{A}_{\mathrm{Pf}} - \hat{A}_{\mathrm{Pf}}^T)$. We parametrize the spin-dependent scalars $\eta$ as node outputs for a fixed number of spin configurations $N_{\mathrm{s}}$. Because the change in spin configuration does not grow with system size, $N_{\mathrm{s}}$ is fixed. We generate two sets of these parameters, on for $\Phi_{\mathrm{Pf}}$ and on for $\tilde{\Phi}_{\mathrm{Pf}}$. App. D provides definitions for the wave function, the MetaGNN, and the parametrization.

**Pretraining.** Previous work like Gao & Günnemann (2023a) needed to canonicalize the Hartree-Fock solutions for different systems before pretraining to ensure that the orbitals fit the neural network. Alternatively, Scherbela et al. (2023) relied on traditional quantum chemistry methods like Foster & Boys (1960)'s localization to canonicalize their orbitals in conjunction with sign equivariant neural networks. In contrast, we ensure that the transformed Hartree-Fock orbitals are similar across structures as we optimize $T \in SO(N_{\mathrm{o}})$ and $A_{\mathrm{HF}} \in \mathbb{S}$ for each structure separately, which simultaneously also accounts for arbitrary rotations in the orbitals produced by Hartree-Fock.

**Limitations.** While our Pfaffian-based generalized wave function significantly improves accuracy on organic chemistry, we leave the transfer to periodic systems for future work (Kosmala et al., 2023). Further, due to the lac of low-level hardware/software support for the Pfaffian and the increased number of orbitals $N_{\mathrm{o}} \geq \max\{N_{\uparrow}, N_{\downarrow}\}$, our Pfaffian is slower than a comparably-sized Slater determinant. While we solve the issue of enforcing the fermionic antisymmetry, our neural wave functions are still unaware of any symmetries of the wave function itself. These are challenging to describe and largely unknown, but their integration may improve generalization performance (Schütt et al., 2018). Finally, in classical single-structure calculations, NeurPf may not improve accuracies. App. P discusses the broader impact of our work.

## 5 Experiments

In the following, we evaluate NeurPf on several atomic and molecular systems by comparing it to Globe (Gao & Günnemann, 2023a) and TAO (Scherbela et al., 2024). Concretely, we investigate the following: (1) Second-row elements and their ionization potentials and electron affinities. Globe cannot compute these due to its restriction to singlet state systems. (2) The challenging nitrogen potential energy surface where Globe significantly degraded performance when enlarging their training set with additional molecules. (3) The TinyMol dataset (Scherbela et al., 2024) to evaluate NeurPf's generalization capabilities across biochemical molecules. In interpreting the following results, one should mind the variational principle, i.e., lower energies are better for neural wave functions. Further, $1 \, \mathrm{kcal} \, \mathrm{mol}^{-1} \approx 1.6 \, \mathrm{m}E_{\mathrm{h}}$ is the typical threshold for chemical accuracy.

Like previous work, we optimize the neural wave function using the VMC framework from Sec. 2. We precondition the gradient with the Spring optimizer (Goldshlager et al., 2024). App. E details the setup further. App. F,I and J show an experiment on extensity and additional ablations.

**Atomic systems and spin configurations.** We evaluate NeurPf on second-row elements and their ionization potentials and electron affinities. These systems are particularly interesting as they represent

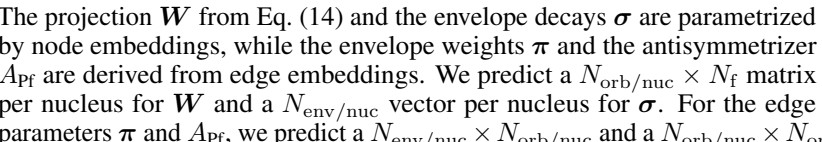

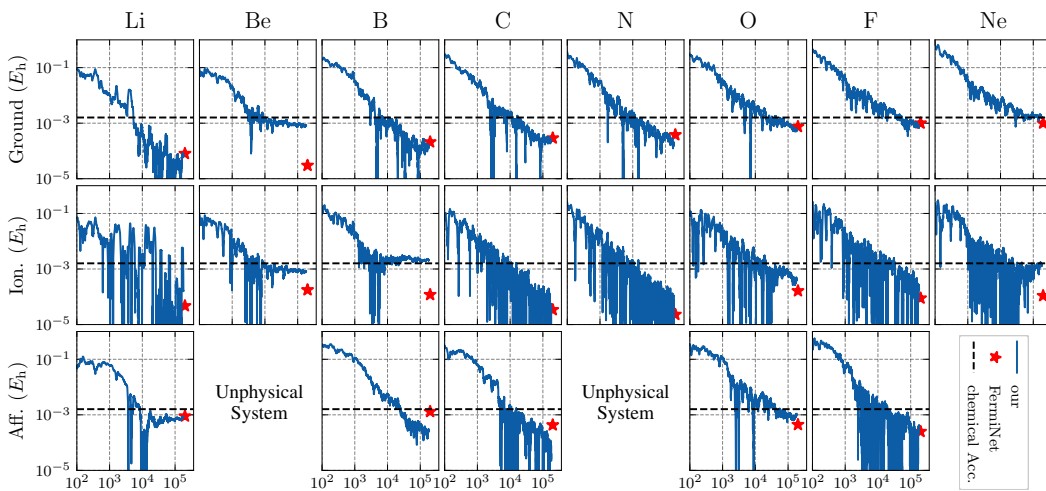

Fig. 3: Ground state, electron affinity, and ionization potential errors of second-row elements during training. A single NeurPf has been trained on all systems jointly while references (Pfau et al., 2020) were calculated separately for each system. Energies are averaged over the last 10% of steps.

a wide range of spin configurations. We cannot use Globe on such systems because they differ from the singlet state assumption. Instead, we compare our results to the single-structure calculations from Pfau et al. (2020)'s FermiNet and the exact results from Chakravorty et al. (1993); Klopper et al. (2010). In App. G, we repeat this experiment for metals.

Fig. 3 displays the ground state energy, electron affinity, and ionization potential errors of NeurPf during training compared to the reference energies from Pfau et al. (2020); Chakravorty et al. (1993); Klopper et al. (2010). It is apparent that NeurPf reaches chemical accuracy relative to the exact results while only training a single neural network for all systems. While separately optimized FermiNets may achieve lower errors, Pfau et al. (2020) trained 21 neural networks for 200k steps each compared to a single NeurPf trained for 200k steps, i.e., 21 times fewer steps and samples. Whereas Gao & Günnemann (2023a); Scherbela et al. (2023) focus on singlet state systems or stable biochemical molecules, NeurPf demonstrates that a generalized wave function need not be restricted to such simple systems and can even generalize to a wide range of electronic configurations.

**Effect of uncorrelated data.** Next, we evaluate NeurPf on the nitrogen potential energy surface, a traditionally challenging system due to its high electron correlation effects (Lyakh et al., 2012). This is particularly interesting as Gao & Günnemann (2023a) observed a significant accuracy degradation when reformulating their wave function to generalize over different systems. In particular, they found that training only on the nitrogen dimer leads to significantly lower errors than training with an ethene-augmented dataset, indicating an accuracy penalty in generalization. We replicate their setup and compare the perfor-

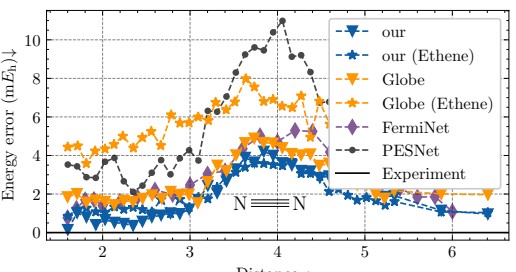

Fig. 4: Potential energy surface of nitrogen. Energies are relative to Le Roy et al. (2006).

mance of NeurPf trained on the nitrogen energy surface with and without additional ethene structures. Like Gao & Günnemann (2023a), the nitrogen structures are taken from Pfau et al. (2020) and the ethene structures from Scherbela et al. (2022). As additional references, we plot Gao & Günnemann (2022)'s PESNet and Fu et al. (2023)'s FermiNet results.

Fig. 4 shows the error potential energy surface relative to the experimental results from Le Roy et al. (2006). NeurPf reduces the average error on the energy surface from Globe's $2.7\,\mathrm{m}E_\mathrm{h}$ to $2\,\mathrm{m}E_\mathrm{h}$ when training solely on nitrogen structures. When adding the ethene structures, Globe's error increases to $5.3\,\mathrm{m}E_\mathrm{h}$ while NeurPf's error stays constant at $2\,\mathrm{m}E_\mathrm{h}$, a lower error than the Globe without the augmented dataset. These results indicate NeurPf's strong capabilities in approximating ground states while allowing for generalization across different systems without a significant loss in accuracy.

**TinyMol dataset.** Finally, we look at learning a generalized wave function over different molecules and structures. We use the TinyMol dataset (Scherbela et al., 2024), consisting of a small and large dataset. The dataset includes 'gold-standard' CCSD(T) CBS energies. The small set consists of 3 molecules with 2 heavy atoms, while the large set covers 4 molecules with 3 heavy atoms. For each molecule, 10 structures are provided. Here, we compare again both Globe (+Moon) and TAO to NeurPf. All models are directly trained on the small and large test sets.

Fig. 5 shows the mean energy difference to CCSD(T) at different stages of the training. We refer to App. K for a per molecule error attribution. It is apparent that NeurPf yields lower errors than the TAO and Globe after at least 500 steps. On the small structures, NeurPf even matches the CCSD(T) baseline after 16k steps and achieves $1.9\,mE_h$ lower energies after 32k steps. Since VMC methods are variational, i.e., lower energies are always better, NeurPf is more accurate than the CCSD(T) CBS reference. Compared to TAO and Globe, NeurPf reports $5.9\,mE_h$ and $11.3\,mE_h$ lower energies, respectively. On the large structures, we observe a similar pattern where we find NeurPf having a 25 times smaller error than TAO during the early stages of training and reaching $21.1\,mE_h$ lower energies after 32k steps – a 6 times lower error compared to the CCSD(T) baseline. Note that since the CCSD(T) (CBS) energies are neither exact nor variational, the true error to the ground state is unknown. Still, we provide additional numbers for a NeurPf trained for 128k steps in App. K. There, we find NeurPf yielding $4.4\,mE_h$ lower energies on the large structures. These results show that a generalized wave function can achieve high accuracy on various molecular structures without pretraining when not relying

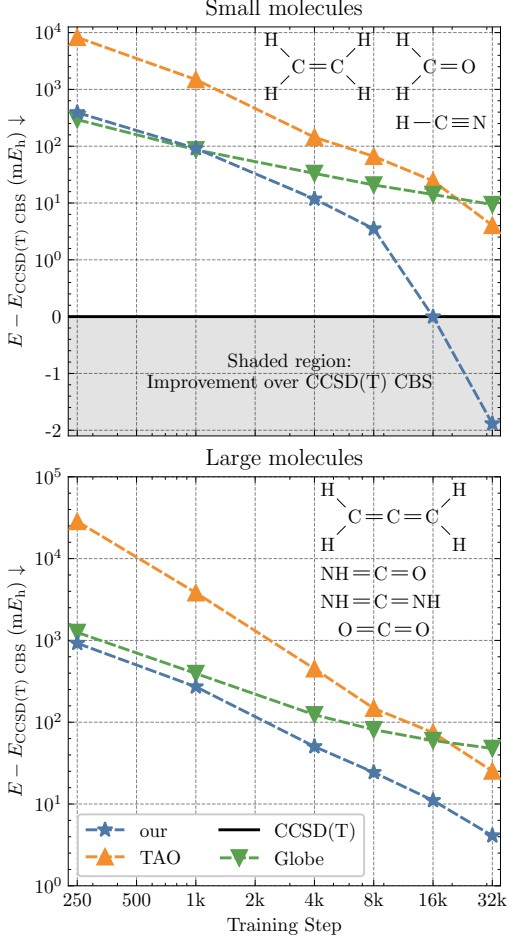

Fig. 5: Convergence of mean energy difference on the TinyMol dataset from Scherbela et al. (2024). The y-axis is linear $< 1$ and logarithmic $\geq 1$. Due to the variational principle, NeurPf is better than the reference CCSD(T) on the small molecules.

on hand-crafted algorithms or Hartree-Fock calculations. For additional experiments, we refer the reader to App. L where we first pretrain TAO and NeurPf on a separate training set and, then, finetune on the small and large test sets and App. M for a comparison of joint and separate optimization.

## 6  Conclusion

In this work, we established a new way of parametrizing neural network wave functions for generalization across molecules via overparametrization with Pfaffians. Our Neural Pfaffian is more accurate, simpler to implement, fully learnable, and applicable to any molecular system compared to previous work. The wave function changes smoothly with the structure, avoiding the discrete orbital selection problem previously solved via hand-crafted algorithms or Hartree-Fock. Additionally, we introduced a memory-efficient implementation of the exponential envelopes, reducing memory requirements while accelerating convergence. Further, we presented a pretraining scheme for Pfaffians enabling initialization with Hartree-Fock – a crucial step for molecular systems. Our experimental evaluation demonstrated that our Neural Pfaffian can generalize across different ionizations of various systems, stay accurate when enlarging datasets, and set a new state of the art by outperforming previous neural wave functions and the reference CCSD(T) CBS on the TinyMol dataset. These developments open the door for new neural wave functions applications, e.g., to generate reference data for machine-learning force fields or density functional theory (Cheng et al., 2024; Gao et al., 2024).

**Acknowledgments.** We greatly thank Simon Geisler for our valuable discussions. Further, we thank Valerie Engelmayer, Leo Schwinn, and Aman Saxena for their invaluable feedback on the manuscript. Funded by the Federal Ministry of Education and Research (BMBF) and the Free State of Bavaria under the Excellence Strategy of the Federal Government and the Länder.

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

Table 1: Number of envelope parameters for the full envelope and our memory efficient envelopes for an explanatory system.

| | $\sigma$ | $\pi$ | Total |
|---|---|---|---|
| full | 1600 | 1600 | 3200 |
| our | 640 | 12800 | 13400 |

## A    Odd numbers of electrons

To handle odd numbers of electrons, we extend the electron pair matrix $\hat{\Phi}_{\mathrm{Pf}} A_{\mathrm{Pf}} \hat{\Phi}_{\mathrm{Pf}}^T$ to even dimensions. We accomplish this by augmenting $\hat{\Phi}_{\mathrm{Pf}} A_{\mathrm{Pf}} \hat{\Phi}_{\mathrm{Pf}}^T$ with a learnable single-electron orbital $\phi_{\mathrm{odd}}$ to

$$\widehat{\hat{\Phi}_{\mathrm{Pf}} A_{\mathrm{Pf}} \hat{\Phi}_{\mathrm{Pf}}^T} = \begin{pmatrix} \hat{\Phi}_{\mathrm{Pf}} A_{\mathrm{Pf}} \hat{\Phi}_{\mathrm{Pf}}^T & \phi_{\mathrm{odd}} \\ -\phi_{\mathrm{odd}}^T & 0 \end{pmatrix}. \tag{21}$$

To obtain a single additional orbital for the whole molecule, we parameterize one orbital $\phi_{\mathrm{odd},m}$ for each nucleus as in Eq. (14) and sum them up to obtain $\phi_{\mathrm{odd}} = \sum_{m=1}^{N_{\mathrm{n}}} \phi_{\mathrm{odd},m}$.

## B    Difference to bottleneck envelopes

Similar to the bottleneck envelope from Pfau et al. (2024), our efficient envelopes aim at reducing memory requirements. The bottleneck envelopes are defined as

$$\chi_{\mathrm{bottleneck}}^k (r_{bi})_j = \sum_{l=1}^L w_{jl}^k \sum_{m=1}^{N_{\mathrm{n}}} \pi_{lm} \exp\left(-\sigma_{lm} \|\boldsymbol{r}_i - \boldsymbol{R}_m\|\right) \tag{22}$$

While both methods share the idea of reducing the number of parameters, they differ in their implementation. Whereas the bottleneck envelopes construct a full set of $L$ many-nuclei envelopes and then linearly recombine these to the final envelopes for each of $K \times N_o$ orbitals, our efficient envelopes construct the final envelopes directly from a set of single-nuclei exponentials. Further, we use a different set of basis functions for each of the $K$ determinants. In terms of computational complexity, the bottleneck envelopes require $O(N_{\mathrm{e}} N_{\mathrm{n}} L) + O(KL N_{\mathrm{e}} N_o)$ operations to compute the envelopes, while our efficient envelopes require $O(K N_{\mathrm{env}} N_{\mathrm{e}} N_o)$ operations. In practice, we found our efficient envelopes to be faster and converge better on all systems we tested. An ablation study is presented in App. I. Further, we observed no numerical instabilities in our envelopes as reported by Pfau et al. (2024).

Compared to the full envelopes, we find our memory efficient ones to be slower but yielding better performance. This is likely due to the increased number of wave function parameters. The number of parameters for the full envelopes and our memory efficient envelopes is shown in Tab. 1 for an example with $N_e = N_o = 20, N_n = 5, N_d = 16, N_{\frac{\mathrm{env}}{\mathrm{atom}}} = 8$. The full envelopes' $\sigma, \pi$ scale both $O(N_d N_n N_o)$ while our memory efficient envelopes' $\sigma$ scales $O(N_d N_n N_{\mathrm{env/nuc}})$ and $\pi$ scales $O(N_d N_n N_{\mathrm{env/nuc}} N_o)$. In runtime, the full envelopes require $O(N_d N_n N_e N_o)$ operations, while our memory efficient envelopes require $O(N_d N_n N_{\frac{\mathrm{env}}{\mathrm{atom}}} N_e N_o)$ operations. In memory complexity, the full envelopes require $O(N_d N_n N_e^2)$, while our memory efficient envelopes require $O(N_d N_n N_{\frac{\mathrm{env}}{\mathrm{atom}}} N_e)$.

## C    Pretraining

To pretrain NeurPf, we solve the optimization problem from Eq. (20). The nested optimization problems are solved iteratively, where we first solve for $T \in SO(N)$ and $A_{\mathrm{HF}} \in \mathbb{S}$ and then for the parameters of the wave function $\theta$. We describe how we parametrize the special orthogonal group $SO(N)$ and the antisymmetric special orthogonal group $\mathbb{S}$ and then how we solve the optimization problems.

To optimize over the special orthogonal group $SO(N)$, we parametrize $T$ via some arbitrary matrix $\tilde{T} \in \mathbb{R}^{N \times N}$. Next, we obtain an antisymmetrized version of $\tilde{T}$ via

$$\hat{T} = \frac{1}{2}\left(\tilde{T} - \tilde{T}^T\right). \tag{23}$$

We now may use $\hat{T}$ with the Cayley transform to obtain a special orthogonal matrix

$$\bar{T} = \left(\hat{T} - I\right)^{-1}\left(\hat{T} + I\right) \tag{24}$$

where $I$ is the identity matrix. $\bar{T}$ is now a special orthogonal matrix where all eigenvalues are 1. To parametrize matrices with an even number of eigenvalues -1 as well, we simply multiply $\bar{T}$ with itself:

$$T = \bar{T}\bar{T} \tag{25}$$

which gives us our final parametrization of the special orthogonal group $SO(N)$ (Gallier, 2013).

We follow Gallier (2013), to parametrize antisymmetric special orthogonal matrices $\mathbb{S}$. In particular, we parametrize some $T$ using the procedure outlined above. To parametrize $A_{\text{HF}}$, it remains to antisymmetrize $T$ while preserving the special orthogonal property. We accomplish this by defining

$$A_{\text{HF}} = T\tilde{I}T^T \tag{26}$$

where

$$\tilde{I} = \text{diag}\left(\begin{bmatrix} 0 & 1 \\ -1 & 0 \end{bmatrix}, \dots\right) = \begin{bmatrix} 0 & 1 & 0 & 0 & \dots \\ -1 & 0 & 0 & 0 & \\ 0 & 0 & 0 & 1 & \\ 0 & 0 & -1 & 0 & \\ \vdots & & & & \ddots \end{bmatrix} \tag{27}$$

is the antisymmetric identity matrix. Since the product of special orthogonal matrices is special orthogonal and $BAB^T$ yielding an antisymmetric matrix for any special orthogonal matrix $B$, we have that $A_{\text{HF}} \in \mathbb{S}$ is an antisymmetric special orthogonal matrix.

Now that we can parametrize both groups with real matrices, we can simplify the optimization problem by performing gradient optimization for both $T$, $A_{\text{HF}}$, and $\theta$. We solve this problem alternatively, where we first solve for $T$ and $A_{\text{HF}}$ by doing $N_{\text{pre}}$ steps of gradient optimization with the prodigy optimizer (Mishchenko & Defazio, 2023) and then perform a single outer step on $\theta$ with the lamb optimizer (You et al., 2020) like previous works (Gao & Günnemann, 2022; von Glehn et al., 2023).

# D   Model architectures

We largely reuse the same architecture for the MetaGNN $\mathcal{M} : (\mathbb{R}^3 \times \mathbb{N}_+)^{N_{\text{n}}} \to \Theta$ and wave function $\Psi_{\text{Pfaffian}} : \mathbb{R}^{N_{\text{e}} \times 3} \times \Theta \to \mathbb{R}$ as Gao & Günnemann (2023a). We canonicalize all molecular structures using the equivariant coordinate frame from Gao & Günnemann (2022).

## D.1   Wave function

Similar to Gao & Günnemann (2023a), we use bars above functions and parameters to indicate that the MetaGNN $\mathcal{M}$ parameterizes these and that they vary by structure. We define our wave function as a Jastrow-Pfaffian wave function like Kim et al. (2023):

$$\Psi(\mathbf{r}) = \exp\left(J(\mathbf{r})\right) \sum_{k=1}^{K} c_k \text{Pf}\left(\hat{\Phi}_{\text{Pf}}^k(\vec{\mathbf{r}}) A_{\text{Pf}}^k \hat{\Phi}_{\text{Pf}}^k(\vec{\mathbf{r}})^T\right). \tag{28}$$

As Jastrow factor $J : \mathbb{R}^{N \times 3} \to \mathbb{R}$ we use a linear combination of a learnable MLP of electron embeddings and the fixed electronic cusp Jastrow from von Glehn et al. (2023):

$$
\begin{aligned}
J(\mathbf{r}) = &\sum_{i=1}^{N} \text{MLP}(\boldsymbol{h}(\vec{r}_i | \vec{\mathbf{r}})) \\
&+ \beta_{\text{par}} \sum_{i,j;\alpha_i = \alpha_j} -\frac{1}{4} \frac{\alpha_{\text{par}}^2}{\alpha_{\text{par}} + \|\boldsymbol{r}_i - \boldsymbol{r}_j\|} \\
&+ \beta_{\text{anti}} \sum_{i,j;\alpha_i \neq \alpha_j} -\frac{1}{2} \frac{\alpha_{\text{anti}}^2}{\alpha_{\text{anti}} + \|\boldsymbol{r}_i - \boldsymbol{r}_j\|}
\end{aligned} \tag{29}
$$

where $\boldsymbol{h} : \mathbb{R}^3 \times \mathbb{R}^{N \times 3} \rightarrow \mathbb{R}^{N_f}$ is the $i$th output of the permutation equivariant neural network, implemented via the Molecular orbital network (Moon) (Gao & Günnemann, 2023a), $\beta_{\text{par}}, \beta_{\text{anti}}, \alpha_{\text{par}}, \alpha_{\text{anti}} \in \mathbb{R}$ are learnable scalars, and $\alpha_i$ is the spin of the $i$th electron.

The orbitals $\hat{\Phi}_{\text{Pf}}$ are defined as in Eq. (14) with Moon performing the following steps: We start with constructing electron embeddings based on electron-electron distances and then proceed to aggregate these embeddings to the orbitals. The nuclei are updated through MLPs and finally diffused to the electrons, yielding the final electron embeddings.

The initial embedding $\boldsymbol{h}_i^{(0)}$ of the $i$th electron is constructed as

$$\boldsymbol{h}_i^{(0)} = \frac{1}{\mu(\boldsymbol{r}_i)} \left( \sum_{j=1}^{N} \sigma \left( \boldsymbol{g}_{ij}^{\text{e-e}} \boldsymbol{W}^{\delta_{\alpha_i}^{\alpha_j}} \right) \circ \Gamma^{\delta_{\alpha_i}^{\alpha_j}} (\|\vec{r}_i - \vec{r}_j\|) \right) \boldsymbol{W} \tag{30}$$

where $\circ$ denotes the Hadamard product and the Kronecker delta $\delta_{\alpha_i}^{\alpha_j}$ as superscript indicates different parameters depending on the identity between spin $\alpha_i$ and $\alpha_j$. $\Gamma : \mathbb{R}^I \rightarrow \mathbb{R}^D$ is a learnable radial filter function, and $\sigma$ is the activation function. $\boldsymbol{g}_{ij}^{\text{e-e}} \in \mathbb{R}^4$ are the rescaled electron-electron distances (von Glehn et al., 2023):

$$\boldsymbol{g}_{ij} = \frac{\log\left(1 + \|\vec{r}_i - \vec{r}_j\|\right)}{\|\vec{r}_i - \vec{r}_j\|} \left[ \vec{r}_i - \vec{r}_j, \|\vec{r}_i - \vec{r}_j\| \right]. \tag{31}$$

$\mu$ is a normalization factor:

$$\mu(\vec{r}) = 1 + \sum_{m=1}^{M} \frac{Z_m}{2} \exp\left( -\frac{\|\vec{r} - \vec{R}_m\|^2}{\sigma_{\text{norm}}^2} \right). \tag{32}$$

We use the initial electron embeddings with nuclei embeddings and electron-nuclei distances to construct pairwise nuclei-electron embeddings representing edges in a fully connected graph:

$$\boldsymbol{h}_{im}^{\text{e-n}} = \sigma \left( \boldsymbol{h}_i^{(0)} + \bar{z}_m + \boldsymbol{g}_{im}^{\text{e-n}} \bar{\boldsymbol{W}}_m \right). \tag{33}$$

where $\bar{z}_m$ is the $m$th nucleus embedding, $\boldsymbol{g}_{im}^{\text{e-n}} \in \mathbb{R}^4$ are the rescaled electron-nuclei distances like in Eq. (31). These embeddings are then aggregated with spatial filters twice: once towards the nuclei and once towards the electrons:

$$\boldsymbol{h}_m^{\text{n}\alpha(1)} = \frac{1}{\mu(\vec{R}_m)} \sum_{i \in \mathbb{A}^\alpha} \boldsymbol{h}_{i,m}^{\text{e-n}} \circ \bar{\Gamma}_m^{\text{n}}(\vec{r}_i - \vec{R}_m), \tag{34}$$

$$\boldsymbol{m}_i^{(1)} = \frac{1}{\mu(\vec{r}_i)} \sum_{m=1}^{M} \boldsymbol{h}_{i,m}^{\text{e-n}} \circ \bar{\Gamma}_m^{\text{e}}(\vec{r}_i - \vec{R}_m), \tag{35}$$

$$\boldsymbol{h}_i^{(1)} = \sigma(\boldsymbol{m}_i^{(1)} \boldsymbol{W} + \boldsymbol{b}). \tag{36}$$

We update the nuclei embeddings with $L$ update layers:

$$\boldsymbol{h}_m^{\text{n}\alpha(l+1)} = \boldsymbol{h}_m^{\text{n}\alpha(l)} + \sigma([\boldsymbol{h}_m^{\text{n}\alpha(l)}, \boldsymbol{h}_m^{\text{n}\hat{\alpha}(l)}] \boldsymbol{W}^{(l)} + \boldsymbol{b}^{(l)}), \tag{37}$$

where $\hat{\alpha}$ denotes the opposite spin of $\alpha$, to obtain the final nuclei embeddings $\boldsymbol{h}_m^{\text{n}\alpha(L)}$. The final electron embeddings $\boldsymbol{h}_i^{\text{e}(L)}$ are constructed by combining the message from the nuclei and the previous electron embedding:

$$\boldsymbol{h}_i^{\text{e}(L)} = \sigma \left( \sigma \left( \boldsymbol{h}_i^{(1)} \boldsymbol{W} + \boldsymbol{m}_i^{(L)} + \boldsymbol{b}_1 \right) \boldsymbol{W} + \boldsymbol{b}_2 \right) + \boldsymbol{h}_i^{(1)} \tag{38}$$

where $\boldsymbol{m}_i$ is the message from the nuclei to the $i$th electron:

$$\boldsymbol{m}_i^{(L)} = \frac{1}{\mu(\vec{r}_i)} \sum_{m=1}^{M} \sigma \left( \left[ \boldsymbol{h}_m^{\text{n}\alpha_i(L)}, \boldsymbol{h}_m^{\text{n}\hat{\alpha}_i(L)} \right] \boldsymbol{W} + \boldsymbol{b} \right) \circ \bar{\Gamma}_m^{\text{diff}}(\vec{r}_i - \vec{R}_m). \tag{39}$$

The spatial filters $\Gamma$ are defined as:

$$\bar{\Gamma}_m^{(l)}(\boldsymbol{x}) = \bar{\beta}_m(\boldsymbol{x})\boldsymbol{W}^{(l)}, \tag{40}$$

$$\bar{\beta}_m(\boldsymbol{x}) = \left[\exp\left(-\left(\frac{\|\boldsymbol{x}\|}{\bar{\varsigma}_{mi}}\right)^2\right)\right]_{i=1}^D \boldsymbol{W}^{\text{env}} \circ \left(\sigma\left(\boldsymbol{x}\bar{\boldsymbol{W}}_m^{(1)} + \bar{\boldsymbol{b}}_m^{(1)}\right)\boldsymbol{W}^{(2)} + \boldsymbol{b}^{(2)}\right). \tag{41}$$

Note that $\bar{\beta}$ is shared across all instances of $\bar{\Gamma}$. $\Gamma$ is defined analogously to $\bar{\Gamma}$ but with fixed learnable parameters instead of MetaGNN parametrized ones.

## D.2 MetaGNN

The MetaGNN $\mathcal{M} : (\mathbb{R}^3 \times \mathbb{N}_+)^{N_n} \to \Theta$ takes the nucleus position $\vec{\mathbf{R}}$ and charges $\mathbf{Z}$ as input and outputs parameters of the electronic wave function to adapt the solution to the system of interest. We follow Gao & Günnemann (2022, 2023a) and implement it as a graph neural network (GNN) where nuclei are represented as nodes and edges are constructed based on inter-particle distances. The charge of the nucleus determines the initial node embeddings:

$$\boldsymbol{k}_i^{(0)} = \boldsymbol{E}_{Z_i} \tag{42}$$

where $\boldsymbol{E}$ is an embedding matrix and $Z_i$ is the charge of the $i$th nucleus. These embeddings are iteratively updated via message passing in the following way:

$$\boldsymbol{k}_i^{(l+1)} = f^{(l)}(\boldsymbol{k}_i^{(l)}, \boldsymbol{t}_i^{(l)}), \tag{43}$$

$$\boldsymbol{t}_i^{(l)} = \frac{1}{\nu_{\vec{R}_i}^{\vec{\mathbf{R}}}} \sum_{j=1}^M g^{(l)}(\boldsymbol{k}_i^{(l)}, \boldsymbol{k}_j^{(l)}) \circ \Gamma^{(l)}(\vec{R}_i - \vec{R}_j), \tag{44}$$

$$\nu_x^{\mathcal{N}} = 1 + \sum_{y \in \mathcal{N}} \exp\left(-\frac{\|x - y\|^2}{\sigma_{\text{norm}}^2}\right) \tag{45}$$

where Eq. (43) describes the update function, Eq. (44) the message construction, and Eq. (45) a learnable normalization coefficient. We implement the functions $f$ and $g$ via Gated Linear Units (GLU) (Shazeer, 2020). As spatial filters, we use the same as in the wave function but additionally multiply the filters with radial Bessel functions from Gasteiger et al. (2019):

$$\Gamma^{(l)}(\boldsymbol{x}) = \beta(\boldsymbol{x})\boldsymbol{W}^{(l)}, \tag{46}$$

$$\beta(\boldsymbol{x}) = \left[\sqrt{\frac{2}{c}}\frac{\sin\left(\frac{f_i x}{c}\right)}{x}\exp\left(-\left(\frac{\|\boldsymbol{x}\|}{\varsigma_i}\right)^2\right)\right]_{i=1}^D \boldsymbol{W}^{\text{env}} \circ \left(\sigma\left(\boldsymbol{x}\boldsymbol{W}^{(1)} + \boldsymbol{b}^{(1)}\right)\boldsymbol{W}^{(2)} + \boldsymbol{b}^{(2)}\right) \tag{47}$$

where $f_i$ are learnable frequencies, and $c$ is a smooth cutoff for the Bessel functions.

After $L$ layers, we take the final node embeddings, pass them through another GLU, and then use a different GLU as head for each distinct parameter tensor of the wave function we want to predict. For edge-dependent parameters, like $\pi$ or $A$, we first construct edge embeddings by concatenating all combinations of node embeddings. We pass these through a GLU and then proceed like for node embeddings. For all outputs, we add a default charge-dependent parameter tensor such that the MetaGNN only learns a delta to an initial guess depending on the charge of the nucleus.

## D.3 Orbital parametrization

Our Pfaffian wave function enables us to simply parametrize a $N_o \geq \max\{N_\uparrow, N_\downarrow\}$ orbitals rather than parametrizing exactly $N_\uparrow/N_\downarrow$. As discussed in Sec. 4.4, we accomplish this by associating a fixed number of orbitals with each nucleus. Here, we provide detailed construction for all parameters of the orbital construction. For simplicity, we do not explicitly show the dependence on the $k$th Pfaffian. Note that we simply extend the readout by an $N_k$ sized dimension for each of the $N_k$ Pfaffians from Eq. (13). Further, we predict two sets of parameters, one for $\Phi_{\text{Pf}}$ and one for $\tilde{\Phi}_{\text{Pf}}$ in

Eq. (9). To parametrize the orbitals, we predict $N_{\mathrm{orb/nuc}}$ orbital parameters for each of the $N_{\mathrm{n}}$ nuclei. Concretely, the linear projection to $\boldsymbol{W}_k$ from Eq. (14) are constructed as

$$\boldsymbol{W} = \begin{bmatrix} \omega_1(\boldsymbol{k}_1) \\ \vdots \\ \omega_{N_{\mathrm{orb/nuc}}}(\boldsymbol{k}_1) \\ \omega_1(\boldsymbol{k}_2) \\ \vdots \\ \omega_{N_{\mathrm{orb/nuc}}}(\boldsymbol{k}_{N_{\mathrm{n}}}) \end{bmatrix} \in \mathbb{R}^{N_{\mathrm{o}} \times N_{\mathrm{f}}} \tag{48}$$

where $\omega_i : \mathbb{R}^D \to \mathbb{R}^{N_{\mathrm{f}}}$ learnable readouts of our MetaGNN. Similarly, we parametrize the envelope coefficients $\boldsymbol{\sigma}_k$ from Eq. (16):

$$\boldsymbol{\sigma} = \begin{bmatrix} \varsigma_1(\boldsymbol{k}_1) \\ \vdots \\ \varsigma_{N_{\mathrm{env/nuc}}}(\boldsymbol{k}_1) \\ \varsigma_1(\boldsymbol{k}_2) \\ \vdots \\ \varsigma_{N_{\mathrm{env/nuc}}}(\boldsymbol{k}_{N_{\mathrm{n}}}) \end{bmatrix} \in \mathbb{R}_+^{N_{\mathrm{env}}} \tag{49}$$

where $\varsigma_i : \mathbb{R}^D \to \mathbb{R}_+$ are learnable readouts of our MetaGNN. The linear orbital weights $\boldsymbol{\pi}$ connect each nuclei-centered envelope to the non-atom-centered orbitals. For this, we need to find a mapping from each of the $N_{\mathrm{env}}$ envelopes to each of the $N_{\mathrm{o}}$ orbitals. Since $N_{\mathrm{env}} = N_{\mathrm{env/nuc}} \times N_{\mathrm{n}}$ and $N_{\mathrm{o}} = N_{\mathrm{orb/nuc}} \times N_{\mathrm{n}}$ are predicted per nuclei, a natural connection is established via a pair-wise atom function:

$$\boldsymbol{\pi} = \begin{bmatrix} \varpi_{1,1}(\boldsymbol{k}_1,\boldsymbol{k}_1) & \cdots & \varpi_{1,N_{\mathrm{orb/nuc}}}(\boldsymbol{k}_1,\boldsymbol{k}_1) & \varpi_{1,1}(\boldsymbol{k}_2,\boldsymbol{k}_1) & \cdots \\ \vdots & \ddots & \vdots & & \\ \varpi_{N_{\mathrm{env/nuc}},1}(\boldsymbol{k}_1,\boldsymbol{k}_1) & \cdots & \varpi_{N_{\mathrm{env/nuc}},N_{\mathrm{orb/nuc}}}(\boldsymbol{k}_1,\boldsymbol{k}_1) & \varpi_{N_{\mathrm{env/nuc}},1}(\boldsymbol{k}_2,\boldsymbol{k}_1) & \cdots \\ \varpi_{1,1}(\boldsymbol{k}_1,\boldsymbol{k}_2) & & \varpi_{1,N_{\mathrm{orb/nuc}}}(\boldsymbol{k}_1,\boldsymbol{k}_2) & \varpi_{1,1}(\boldsymbol{k}_2,\boldsymbol{k}_2) & \cdots \\ \vdots & & \vdots & \vdots & \ddots \end{bmatrix} \in \mathbb{R}^{N_{\mathrm{env}} \times N_{\mathrm{o}}} \tag{50}$$

where $\varpi_{i,j} : \mathbb{R}^D \times \mathbb{R}^D \to \mathbb{R}$ are learnable readouts of our MetaGNN. Similarly, we establish the orbital correlations $A$ from Eq. (14) by connecting each of the $N_{\mathrm{o}}$ orbitals to each other:

$$\hat{A}_{\mathrm{Pf}} = \begin{bmatrix} \alpha_{1,1}(\boldsymbol{k}_1,\boldsymbol{k}_1) & \cdots & \alpha_{1,N_{\mathrm{orb/nuc}}}(\boldsymbol{k}_1,\boldsymbol{k}_2) & \alpha_{1,1}(\boldsymbol{k}_2,\boldsymbol{k}_1) & \cdots \\ \vdots & \ddots & \vdots & & \\ \alpha_{N_{\mathrm{orb/nuc}},1}(\boldsymbol{k}_1,\boldsymbol{k}_1) & \cdots & \alpha_{N_{\mathrm{orb/nuc}},N_{\mathrm{orb/nuc}}}(\boldsymbol{k}_1,\boldsymbol{k}_1) & \alpha_{N_{\mathrm{orb/nuc}},1}(\boldsymbol{k}_2,\boldsymbol{k}_1) & \cdots \\ \alpha_{1,1}(\boldsymbol{k}_1,\boldsymbol{k}_2) & & \alpha_{1,N_{\mathrm{orb/nuc}}}(\boldsymbol{k}_1,\boldsymbol{k}_2) & \alpha_{1,1}(\boldsymbol{k}_2,\boldsymbol{k}_2) & \cdots \\ \vdots & & \vdots & \vdots & \ddots \end{bmatrix} \in \mathbb{R}^{N_{\mathrm{o}} \times N_{\mathrm{o}}} \tag{51}$$

$$A_{\mathrm{Pf}} = \frac{1}{2}(\hat{A}_{\mathrm{Pf}} - \hat{A}_{\mathrm{Pf}}^T) \tag{52}$$

where $\alpha_{i,j} : \mathbb{R}^D \times \mathbb{R}^D \to \mathbb{R}$ are learnable readouts of our MetaGNN and Eq. (52) enforcing the antisymmetry requirements on $A$.

### D.4 Changes to the MetaGNN

We performed several optimizations on the MetaGNN from Gao & Günnemann (2023a) that primarily reduce the number of parameters while keeping accuracy. In particular, we changed the following:

- We replace all MLPs with gated linear units (GLU) (Shazeer, 2020).
- We reduced the hidden dimension from 128 to 64.

Table 2: Hyperparameters used for the experiments.

| | Hyperparameter | Value |
|---|---|---|
| | Structure batch size | full batch |
| | Total electron samples | 4096 |
| **Pretraining** | Epochs | 10000 |
| | Learning rate | $10^{-3} * (1 + t * 10^{-4})^{-1}$ |
| | Optimizer | Lamb |
| | MCMC steps | 5 |
| | Basis | STO-6G |
| | Subproblem steps | 50 |
| | Subproblem optimizer | Prodigy |
| | Subproblem $\alpha$ | 1.0 |
| | Subproblem $\beta$ | $10^{-4}$ |
| **Optimization** | Steps | 60000 |
| | Learning rate | $0.02 * (1 + t * 10^{-4})^{-1}$ |
| | Optimizer | Spring |
| | MCMC steps | 20 |
| | Norm constraint | $10^{-3}$ |
| | Damping | 0.001 |
| | Momentum | 0.99 |
| | Energy clipping | 5 times mean deviation from median |
| **Ansatz** | Hidden dim | 256 |
| | E-E int dim | 32 |
| | Layers | 4 |
| | Activation | SiLU |
| | Determinants/Pfaffians | 16 |
| | Jastrow layers | 3 |
| | Filter hidden dims | $[16, 8]$ |
| **Pfaffian** | $N_{\mathrm{orb/nuc}}$ (H, He) | 2 |
| | $N_{\mathrm{orb/nuc}}$ (Li, Be) | 6 |
| | $N_{\mathrm{orb/nuc}}$ (B, C) | 7 |
| | $N_{\mathrm{orb/nuc}}$ (N, O) | 8 |
| | $N_{\mathrm{orb/nuc}}$ (F, Ne) | 10 |
| | $N_{\mathrm{env/nuc}}$ | 8 |
| **MetaGNN** | Embedding dim | 64 |
| | Message dim | 32 |
| | Layers | 3 |
| | Activation | SiLU |
| | Filter hidden dims | $[32, 16]$ |

- We reduced the message dimension from 64 to 32.

- We use bessel basis functions (Gasteiger et al., 2019) on the radius for edge filters.

- We remove the hand-crafted orbital locations and the associated network.

- We added a LayerNorm before every GLU.

Together, these changes reduce the number of parameters from 13M to 1M for the MetaGNN while outperforming Gao & Günnemann (2023a) as demonstrated in Sec. 5.

# E    Experimental setup

Table 3: Compute time per experiment measured in Nvidia A100 GPU hours.

| Experiment | Time (GPU hours) |
|---|---|
| Ionization & affinity | 224 |
| N2 | 116 |
| N2 + Ethene | 124 |
| TinyMol small | 78 |
| TinyMol large | 96 |

## E.1 Hyperparameters

We list the default parameters used for the experiments in Tab. 2. Most of them were taken directly from Gao & Günnemann (2023a). We may have used different parameters for the experiments in Sec. 5 if explicitly stated so. We implement everything in JAX (Bradbury et al., 2018). To compute the laplacian $\nabla^2 \Psi$, we use the forward laplacian algorithm (Li et al., 2024) implemented in the folx library (Gao et al., 2023).

## E.2 Source code

We provide the source code publicly on GitHub [1].

## E.3 Compute time

Tab. 3 lists the compute times required for conducting our experiments measured in Nvidia A100 GPU hours. Depending on the experiment, we use between 1 and 4 GPUs per experiment via data parallelism. We typically allocated 32GB of system memory and 16 CPU cores per experiment. In terms of the number of parameters, the Moon wave function is as large as in Gao & Günnemann (2023a) at 1M parameters, and the MetaGNN shrank from 13M parameters to just 1M parameters.

## E.4 Preconditioning

The Spring optimizer (Goldshlager et al., 2024) is a natural gradient descent optimizer for electronic wave functions $\Psi$ with the following update rule

$$\theta^t = \theta^t - \eta \delta^t \tag{53}$$

$$\delta^t = (\bar{O}^T \bar{O} + \lambda I)^{-1} (\nabla \theta^t + \lambda \mu \delta^{t-1}) \tag{54}$$

where $\lambda$ is the damping factor, $\mu$ is the momentum, $\eta$ is the learning rate, and $\bar{O}$ is the zero-centered Jacobian:

$$\bar{O} = O - \frac{1}{N} \sum_{i=1}^{N} O_i, \tag{55}$$

$$O = \begin{bmatrix} \frac{\partial \log \psi(x_1)}{\partial \theta} \\ \vdots \\ \frac{\partial \log \psi(x_N)}{\partial \theta} \end{bmatrix}. \tag{56}$$

Since $\bar{O} \in \mathbb{R}^{N \times P}$ where $N$ is the batch size and $P$ the number of parameters, the update in Eq. (54) can be efficiently computed using the Woodbury matrix identity, which after some simplifications yields

$$\delta^t = \bar{O}(\bar{O}\bar{O}^T + \lambda I)^{-1}(\epsilon + \mu\bar{O}\delta^{t-1}) + \mu\delta^{t-1}. \tag{57}$$

Our early experiment found it necessary to center the jacobian $\bar{O}$ per molecule rather than once for all. In single-structure VMC, the centering eliminates the gradient of the wave function along the direction where the amplitude of the wave function increases for all inputs. This direction does not

---

[1] https://github.com/n-gao/neural-pfaffian

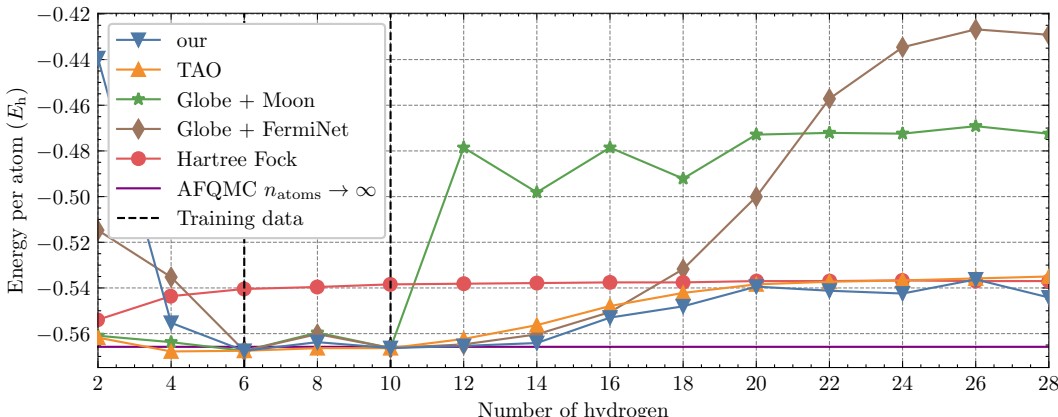

Fig. 6: Energy per atom of hydrogen chains with different lengths. The energy is computed with a single NeurPf trained on the hydrogen chains with 6 and 10 atoms.

affect energies. Thus, instead of restricting the gradient from increasing in magnitude for all samples, we constrain it to not increase in magnitude for each molecule separately N.ote that the latter implies the first but not vice versa. For multi-structure VMC, we compute $\bar{O}$ as

$$\bar{O} = O - \begin{bmatrix} \frac{1}{N_1} \sum_{i=1}^{N_1} O_i \\ \vdots \\ \frac{1}{N_M} \sum_{i=N-N_M}^{N_M} O_i \end{bmatrix} \tag{58}$$

where $N_1, ..., N_M$ are the index limits between molecular structures.

To stabilize computations, we performed preconditioning in float64.

## F    Extensivity on hydrogen chains

Gao & Günnemann (2023a) and Scherbela et al. (2024) analyzed the behavior of their wave functions on hydrogen chains to investigate the extensivity of their wave functions. They did so by training the generalized wave functions on a set of hydrogen chains with 6 and 10 elements. Then, they evaluated the energy per atom on hydrogen chains with different lengths. We replicated their experiment and trained a single NeurPf on the hydrogen chains with 6 and 10 atoms and evaluated the energy per atom on hydrogen chains of increasing lengths.

Fig. 6 shows the energy per atom of hydrogen chains with different lengths for various methods, Globe+Moon and Globe+FermiNet from Gao & Günnemann (2023a), Scherbela et al. (2024), Hartree-Fock (CBS), the AFQMC limit for an infinitely long chain (Motta et al., 2017), and NeurPf. It is apparent that NeurPf outperforms Globe+Moon and Globe+FermiNet significantly by achieving significantly lower energies outside of the training regime. Compared to Scherbela et al. (2024), NeurPf generally performs better on longer chains, achieving errors below the Hartree-Fock baseline. However, we observe significantly higher errors in the shortest chains in NeurPf.

These results indicate that NeurPf is better at generalizing to longer chains than previous works despite not including additional Hartree-Fock calculations like Scherbela et al. (2024).

## G    Metal ionization energies

In addition to the results in Sec. 5, where we train on all second-row elements and their ionization and affinity potentials, we here train a single NeurPf on a set of metals and their ionization energies. This demonstrates that Neural Pfaffians also scale to heavier 3rd and 4th row elements. Fig. 7 shows the ionization energy during training. It is apparent, that NeurPf can learn a solution for all states simultaneously.

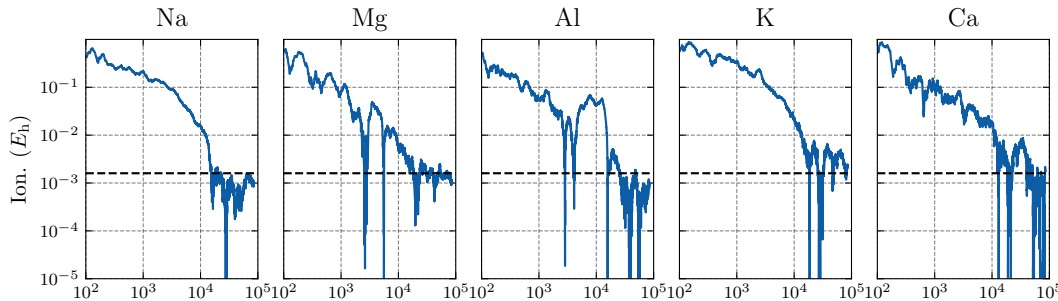

Fig. 7: Ionization energies of metal atoms. The ionization energies are computed with a single NeurPf trained on the neutral and ionized atoms. Reference energies are taken from Martin & Musgrove (1998).

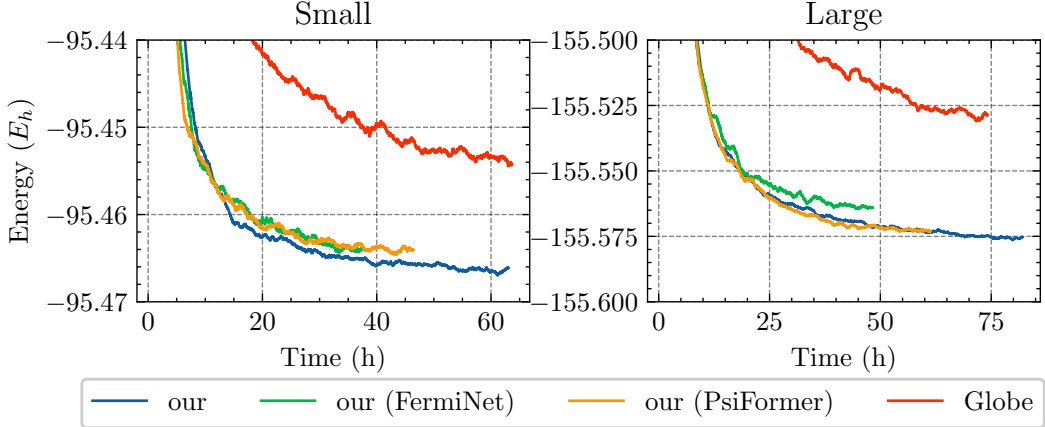

Fig. 8: Energy convergence as a function of time.

## H  TinyMol convergence in time

In Fig. 8, we show the runtime effect of choosing different embedding and antisymmetrizer. We test our default model, our (FermiNet), our (PsiFormer) and Globe + Moon on both TinyMol datasets. For any time budget, all variants of NeurPf converge to lower energies than Globe.

## I  Convergence ablation studies

Here, we provide additional ablation studies to further investigate the performance of NeurPf and our efficient envelopes. In particular, we train four different models on the small TinyMol dataset: NeurPf, NeurPf with the envelopes from Spencer et al. (2020), NeurPf with the envelopes from Pfau et al. (2024), and an AGP-based generalized wave function.

The total energy during training is shown in Fig. 9. The left plot shows the convergence regarding the number of steps, and the right plot shows the convergence in terms of time. We observe that NeurPf convergence is consistently faster than the other methods in terms of the number of steps and time. One further sees the importance of generalizing Eq. (9) via the Pfaffian as the AGP-based wave function does not converge to the same accuracy as NeurPf. The bottleneck envelopes from Pfau et al. (2024) do not only converge to worse energies but are also slower per step than our efficient envelopes from Sec. 4.2.

## J  Model ablation studies

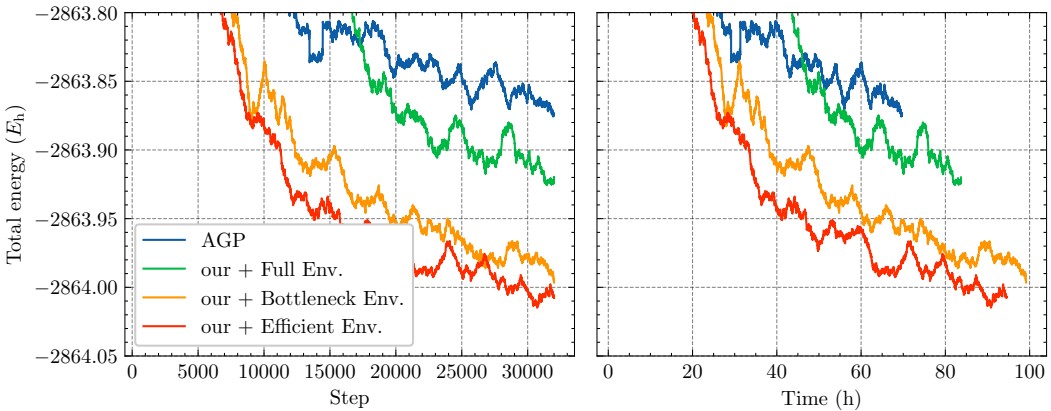

Fig. 9: Ablation study on the small TinyMol dataset. The y-axis shows the sum of all energies in the dataset. The left plot shows the convergence in terms of the number of steps. The right plot shows the convergence in terms of time. our + Full Env. shows a NeurPf with the envelopes from Spencer et al. (2020) and our + Bottleneck Env. uses the bottleneck envelopes from Pfau et al. (2024).

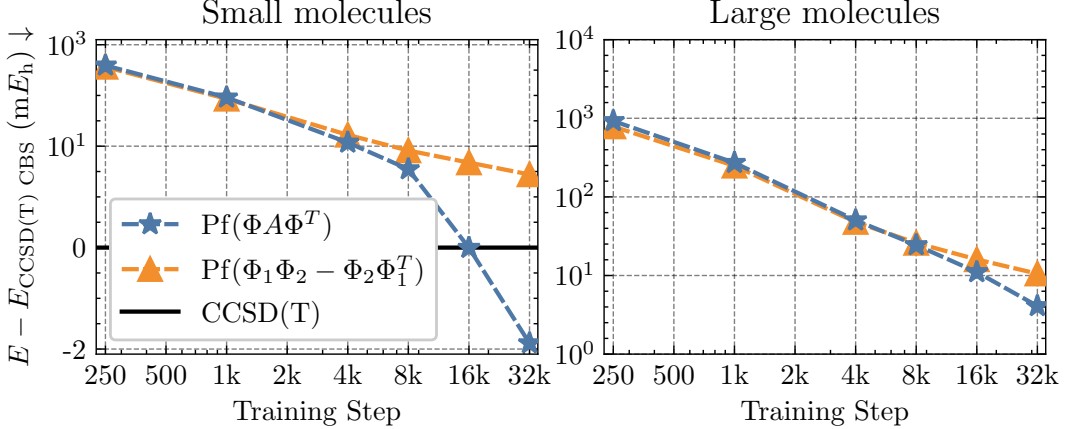

Fig. 10: TinyMol ablation with fixed and learnable antisymmetrizer.

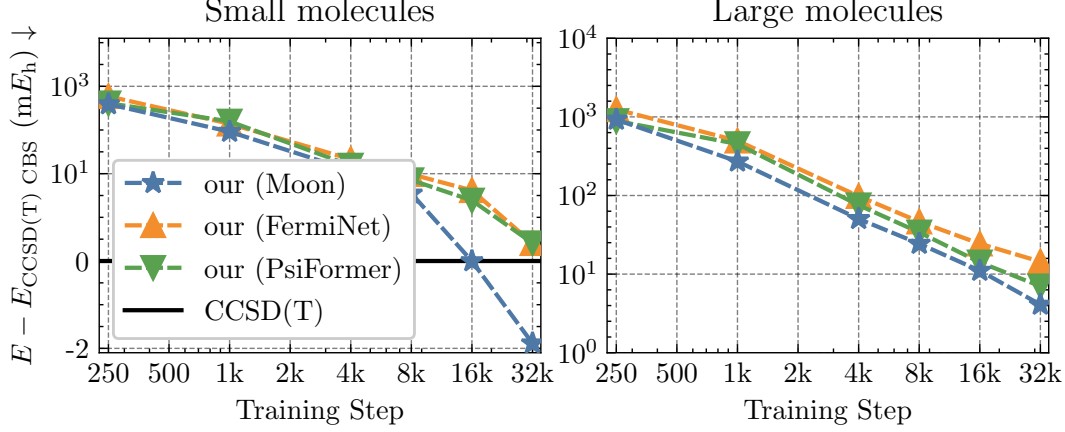

Fig. 11: Ablation study on the small TinyMol dataset with different embedding networks.

Table 4: TinyMol energies compared to CCSD(T) in m$E_h$.

| Method (Steps) | Small | | | Large | | | |
|---|---|---|---|---|---|---|---|
| | CNH | $C_2H_4$ | $COH_2$ | $C_3H_4$ | $CN_2H_2$ | CNOH | $CO_2$ |
| Globe (32k) | 5.2 | 12.3 | 10.7 | 62.3 | 45.8 | 40.4 | 42.7 |
| TAO (32k) | 1.1 | 4.5 | 6.6 | 18.7 | 21.0 | 41.9 | 19.6 |
| our (32k) | **-3.7** | **0.1** | **-2.1** | **12.7** | **5.5** | **3.1** | **5.0** |
| our (128k) | -4.2 | -1.5 | -3.7 | 1.4 | -3.8 | -6.9 | -8.2 |

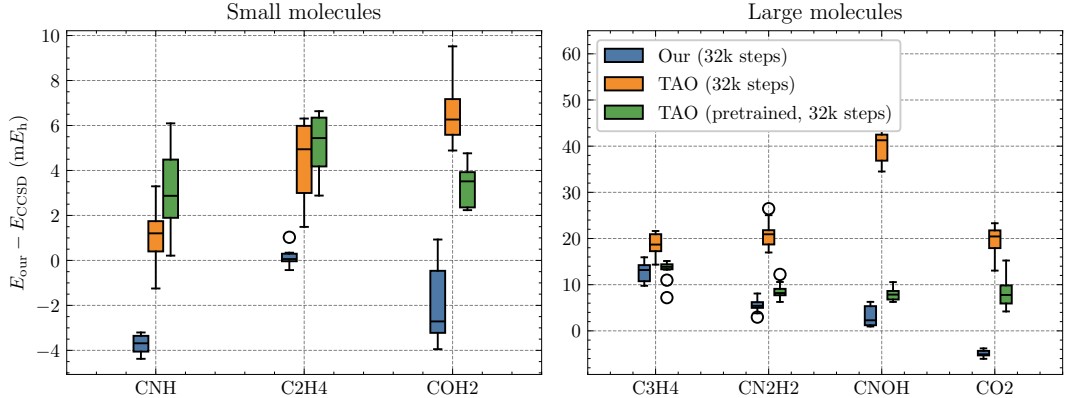

Fig. 12: Boxplot of the energy per molecule on both TinyMol small and large datasets for NeurPf, TAO, and the pretrained TAO from Scherbela et al. (2024). Each boxplot contains results from 10 structures for the given molecule. The line indicates the mean, the box the interquartile range, and the whiskers the 1.5 times the interquartile range.

## J.1 Learnable antisymmetrizer

We picked $Pf(\Phi A \Phi^T)$ as parametrization because it generalizes Slater determinants and many alternative parametrizations. For instance, by choosing $A = \begin{pmatrix} 0 & I \\ -I & 0 \end{pmatrix}$ and $\Phi = (\Phi_1 \ \Phi_2) = Pf(\Phi A \Phi^T) \implies Pf(\Phi_1 \Phi_2^T - \Phi_2 \Phi_1^T)$. We investigate the impact of having $A$ being fixed/learnable in Fig. 10. The results suggest that having $A$ being learnable is a significant factor in our Neural Pfaffian's accuracy.

## J.2 Embedding network

Since NeurPf is not limited to Moon, we performed additional ablations with FermiNet (Pfau et al., 2020) and PsiFormer (von Glehn et al., 2023) as the embedding. The results in Fig. 11 show Neural Pfaffians outperforming Globe and TAO with any of the three equivariant embedding models. Consistent with Gao & Günnemann (2023a), Moon is the best choice for generalized wave functions.

# K TinyMol results

Here, we provide additional data analysis and error metrics for the TinyMol dataset. First, we show in Table 4 the energy per molecule for the small and large TinyMol datasets for NeurPf, Globe, and TAO. To estimate the remaining error, we also train another NeurPf for 128k steps. The results show that NeurPf consistently outperforms TAO and Globe on all molecules in both datasets.

Second, we show the error per molecule for both the small and large TinyMol datasets in Fig. 12. We plot all models after 32k steps of training. It is apparent that NeurPf consistently results in lower, i.e., better, energies than TAO on all molecules in both datasets. Even the pretrained TAO is outperformed by NeurPf on all but four structures of C3H4 in the large TinyMol dataset.

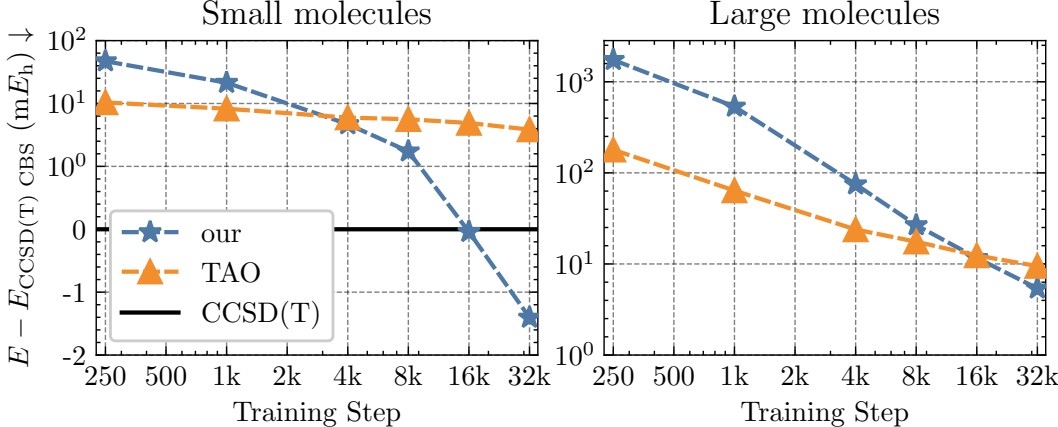

Fig. 13: TinyMol results with pretraining on the training set.

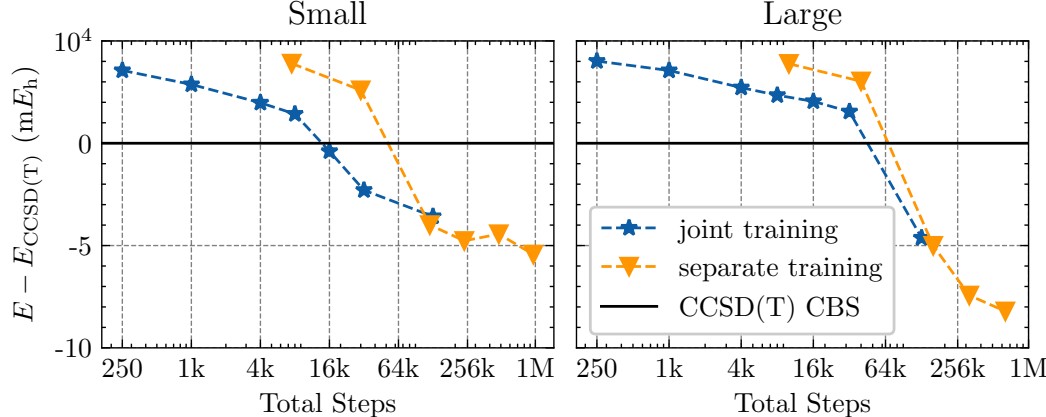

Fig. 14: Comparison of the energy per molecule on the TinyMol dataset for training jointly on all structures vs training a model per structure.

## L   Pretraining on the TinyMol dataset

The TinyMol provides an additional pretraining set of 360 structures (18 molecules, 20 structures each). Like Scherbela et al. (2024), we pretrain our model on the training set of the TinyMol dataset and then finetune on the two test sets. Interestingly, we find the Spring optimizer to be unstable when swapping molecules from step to step and, thus, use CG-preconditioning like Gao & Günnemann (2023a) during pretraining. While yielding a small benefit on the small molecules, we find no notable difference to the Hartree-Fock pretrained model on the large molecules as shown in Fig. 13. On the small structures, the unpretrained NeurPf's energies are $5.7\,\mathrm{m}E_{\mathrm{h}}$ lower. NeurPf also surpasses the pretrained TAO after just 8k steps. Compared to the pretrained TAO on the large structures, NeurPf surpasses TAO after 16k steps and achieves $5.4\,\mathrm{m}E_{\mathrm{h}}$ lower energies after 32k steps.

## M   Joint vs separate training

To estimate the benefit of training a generalized wave function compared to training a model per molecule, we compare the convergence of the total energy on the TinyMol dataset for both approaches depending on the total number of training steps. As training a separate model for each of the 70 TinyMole test molecules is computationally beyond the scope of this work, we select on structure per molecules and train a model for each of the 7 molecules. We use the same NeurPf with MetaGNN for both approaches. The results are shown in Fig. 14. We observe that for lower step numbers, it is quite beneficial to train a generalized model. Though, this benefit vanishes for higher step numbers, and

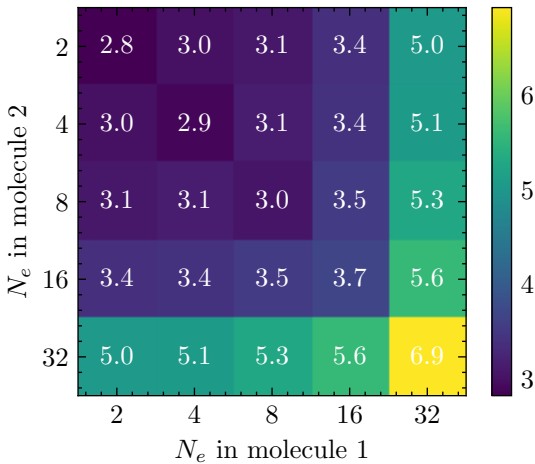

Fig. 15: Time per training step depending on the number of electrons in two molecules.

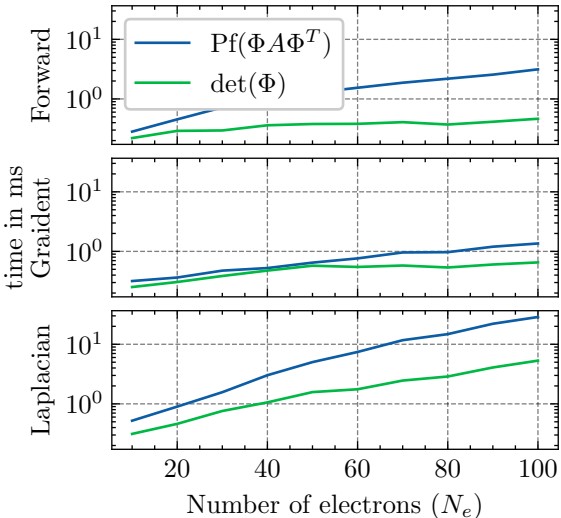

Fig. 16: Time for the forward pass, gradient and Laplacian computation of determinant vs our Pfaffian implementation.

training a model per molecule yields lower energies. We attribute this to the fact that the generalized model has to learn a more complex representation that is not necessary for each molecule individually. Further, the per-molecule energy estimates are quite unstable due to the small shared batch size. Developments like Scherbela et al. (2023) may improve NeurPf training as well.

## N  Training time by batch composition

Here, we benchmark the total time per step for a two-molecule batch. We test all combinations of two molecules with $N_e^1, N_e^2 \in 2, 4, 8, 16, 32$. While we find a small runtime increase when processing small molecules jointly in Fig. 15, for larger systems, we see the runtime per step converge to the geometric mean of the individual runtimes.

## O  Pfaffian runtime

In Fig. 16, we benchmark our implementation for $\mathrm{Pf}(\Phi A \Phi^T)$ (incl. the matrix multiplications) against the standard operation of $\det \Phi$ for 10 to 100 electrons. We implement the Pfaffian in JAX

while highly optimized CUDA kernels are available for the determinant. In summary, both share the same complexity of $O(N^3)$, but the Pfaffian is approximately 5 times slower.

## P   Broader impact

Highly accurate quantum chemical calculations are essential for understanding chemical reactions and materials properties. Our work contributes to this development by providing accurate neural network quantum Monte Carlo calculations at broader scales thanks to generalized wave functions. While this may be used to distill more accurate force fields or exchange-correlation functionals for DFT, the societal impact of our work is primarily in the scientific domain due to the high computational cost of neural network VMC. To the best of our knowledge, our work does not promote any negative societal impact more than general theoretical chemistry research does.

