# OpenReview forum: "Neural Pfaffians: Solving Many Many-Electron Schrödinger Equations"
_NeurIPS.cc/2024/Conference — NeurIPS 2024 oral_

### Official Review · Reviewer_F2V3 · 2024-07-08

**Soundness:** 2
**Presentation:** 3
**Contribution:** 3
**Rating:** 6
**Confidence:** 3

**Summary:**

In this paper, the authors propose a novel Pfafifian-based anti-symmetrization method to represent the generalized wavefunction in quantum chemistry. Unlike the traditional Slater determinant-based anti-symmetrization, the Pfaffian-based method offers greater flexibility in selecting the number of orbitals. This flexibility is crucial for generating a universal wavefunction representation for molecular systems with varying number of electrons. The authors also introduce a new pretraining scheme that addresses the rotational symmetry in Hartree-Fock (HF) solutions, thereby improving the quality of the initial guess in their method. They validate the effectiveness of their approach through experiments on atomic systems and the TinyMol dataset.

**Strengths:**

1.	To the review's knowledge, this is the first study that uses Pfaffian as an anti-symmetrization method in the area of solving the Schrödinger equation with neural networks.
2.	The proposed pretraining scheme effectively avoids orbital disorder in HF solutions, ensuring that the initial guess for the wavefunction is more accurate and stable. This scheme can be integrated with the other methods in solving the Schrödinger equation, facilitating further development and refinement in this area.
3.	The performance of the proposed method is not affected by the uncorrelated data, indicating its potential to incorporate more training data to improve the transferability of the proposed method.

**Weaknesses:**

1.	The primary techniques in this paper are the Pfaffian anti-symmetrization method and the associated pretraining scheme. However, in the TinyMol dataset evaluation, the equivariant part of the proposed method differs from that of the baseline, making it difficult to attribute performance difference solely to the new anti-symmetrization method. Additionally, the performance of the proposed method is not consistently superior to the baseline. The reviewer recommends an ablation study to clearly demonstrate the effectiveness of the proposed method.
2.	To handle odd numbers of electrons, the authors introduce a learnable padding vector to prevent Pfaffian collapse. While this approach appears effective in in-distribution experiments (e.g., affinity and ionization potential of atomic systems), the reviewer is concerned about its efficacy in out-of-distribution scenarios, particularly when generalizing to unseen systems with different electron numbers.

**Questions:**

There are other ways to construct a skew-symmetric matrix, such as $\Phi_1^{\top}\Phi_2-\Phi_2^{\top}\Phi_1$, where $\Phi_1,\Phi_2\in\mathbb{R}^{N_{0}\times N}$ are the output of different equivariant networks. The reviewer is curious about why the authors choose $\Phi^{\top}A\Phi$ in Neural Pfaffians.

**Limitations:**

See weakness part

---

> ### Author Rebuttal · Authors · 2024-08-06
>
> We thank the reviewer for their invaluable feedback and hope to address their concerns. Firstly, we would like to highlight the broad range of new experimental evidence we present in the general comment. The following details how the experiments relate to the reviewer's concerns.
>
> **Ablation studies**\
> To better isolate the contribution of our Neural Pfaffians, we performed several new ablation studies on the TinyMol dataset. In particular, we also trained Globe (+ Moon) [1] on the TinyMol dataset and combined our Neural Pfaffians with FermiNet [2] and PsiFormer [3].
>
> Globe shares the same embedding technique as our Neural Pfaffian and, thus, only differs in how to enforce the fermionic antisymmetry. The results of Globe are depicted in Figure 1 of the general response. There, we can see that while Globe starts similar to our Neural Pfaffians, it cannot reach similar accuracies and converges to significantly higher, i.e., worse, energies.
>
> Since the Neural Pfaffian is not restricted to Moon as an embedding technique, we perform an ablation study where we replace Moon with FemriNet and PsiFormer, respectively. The convergence plots are depicted in Figure 2 of the general response. There, we can see that our Neural Pfaffian outperforms TAO and Globe, independent of the choice of embedding model. However, consistent with [1], Moon performs better in generalized wave functions.
>
> Additionally, we would like to point the reviewer to the ablation study in Appendix F, where we replace the Pfaffians with AGPs $\Psi=\det(\Phi_\uparrow\Phi_\downarrow^T)$, a special case of the Pfaffian that is faster to evaluate. The rest of the network stays the same. There, we find that AGPs are significantly faster to compute, though they cannot match the accuracy of our Neural Pfaffians.
>
> **Comparison to CCSD(T)**\
> Our NeurPf does not match the CCSD(T) energies in Figure 5 of the paper, as convergence typically requires 100k-200k steps [2,3]. We only trained for 32k steps to match the setup from [4]. To better illustrate final convergence, we added an evaluation to Table 1 of the general response, where we train a NeurPf for 128k steps. Our NeurPf comes within chemical accuracy ($\leq 1.6mE_h$) or surpasses CCSD(T) even on the larger dataset.
>
> **Odd numbers of electrons**\
> We are happy to present an alternative solution to deal with odd numbers of electrons in the general comment and would appreciate the reviewer's opinion on this. In short, instead of appending a learnable vector to the orbital matrix, we pad the orbitals $\Phi$ in both dimensions with an identity block to obtain $\hat{\Phi}=\begin{pmatrix}\Phi&0\\\\0&1\end{pmatrix}$. Additionally, we also pad the antisymmetrizer $A$ to $\hat{A}=\begin{pmatrix}A&1\\\\-1&0\end{pmatrix}$ such that one obtains $\text{Pf}(\hat{\Phi} \hat{A}\hat{\Phi}^T)\propto\det\Phi$ if $\Phi$ is square.
>
> We repeat the experiment with this new formulation on the second-row elements experiment in Figure 6 of the general comment. We find little difference. Going forward, we will adopt the new padding technique as it requires no additional parameters.
>
> **Construction of skew-symmetric matrix**\
> This is a great point raised by the reviewer. We agree with the reviewer that there are various ways of parametrizing skew-symmetric matrices. We decided to go with $\Phi A\Phi^T$ as a general construction. For instance, $A=\begin{pmatrix}0&I\\\\ -I&0\end{pmatrix}$ and $\Phi=(\Phi_1 \hspace{1em} \Phi_2)$ corresponds to the reviewer's suggestion $\Phi A\Phi^T=\Phi_1\Phi_2^T - \Phi_2\Phi_1^T$. We experimentally verify the advantage of having $A$ being fully learnable in Figure 3 of the general response, where we compare our approach to a fixed non-learnable $A$.
>
> The results indicate that having $A$ as learnable grants an accuracy benefit in the later stages of training. During the development of our method, we experimented with several other parametrizations, e.g., $A$ being learnable block-diagonal or other fixed forms like $A=\text{diag}\left(\begin{pmatrix}0&I\\\\-I&0\end{pmatrix}, ...\right)$. Still, all of these resulted in very similar training trajectories to the one depicted in Figure 3 of the general response. We also experimented with parametrizing the skew-symmetric matrix for the Pfaffian directly via pair-orbitals $\text{Pf}(A)$ with $A_{ij}=\phi(h_i, h_j) - \phi(h_j, h_i)$ but found this to be numerically unstable for molecular systems, especially with heavier atoms. To better communicate this, we will add this experiment among all the other new experimental evidence to the paper.
>
> **Final remarks**\
> We hope to have addressed the reviewer's concerns and were able to isolate our contribution better with our additional experimental evidence. We are happy to discuss further concerns and look forward to an engaging discussion.
>
> [1] Gao et al. "Generalizing Neural Wave Functions"\
> [2] Pfau et al. "Ab-Initio Solution of the Many-Electron Schrödinger Equation with Deep Neural Networks"\
> [3] von Glehn et al. "A Self-Attention Ansatz for Ab-initio Quantum Chemistry"\
> [4] Scherbela et al. "Towards a transferable fermionic neural wavefunction for molecules"

---

> > ### Comment · Reviewer_F2V3 · 2024-08-09
> >
> > Thank you for your response. It has adequately addressed my concerns, so I'd like to increase the score to 6.

---

### Official Review · Reviewer_3Wr9 · 2024-07-09

**Soundness:** 4
**Presentation:** 3
**Contribution:** 4
**Rating:** 8
**Confidence:** 3

**Summary:**

This paper proposes a new ansatz (Neural Phaffian) for parameterizing wave functions. The new ansatz improves the expressive power by making it possible to increase the number of orbitals. It is also beneficial to tasks like generalization between different systems. The effectiveness is demonstrated with plenty of experiments.

**Strengths:**

-	The paper targets important tasks in physical sciences. It is laudable that the authors not only address ground-state energy calculation but also consider ionization energy and generalization among different systems, which are of more practical significance.
-	Handling anti-symmetry with Pfaffian is a very novel and clever idea! How to enforce anti-symmetry is one of the most important problems in this task, and for decades we have not been so far from original Slater determinants. This work provides rich insights and opens up great opportunities for future research.
-	The authors present comprehensive empirical studies to demonstrate the advantages of their model.

**Weaknesses:**

-	As mentioned in L250-258, Neural Pfaffian is slower than a comparably-sized wave function with Slater determinants. It would be better if the authors could add training-convergence plots with time instead of iterations when comparing Neural Pfaffian with baseline methods.
-	The metric of summing over all energies in a dataset, as in Fig 7, is quite weird. I do not think this metric makes much sense. Additionally, it is possible that some systems with large absolute energy values dominate the curve. It is not convincing enough to showcase the advantage in **most** systems in the dataset.
-	There are several typos. I highly recommend the authors read the paper thoroughly again to fix all the typos. Just to list some of them:
1.	L23: It should be $ \langle \Psi | \hat{H} | \Psi \rangle$.
2.	L120: ‘inferring’.

**Questions:**

1.	Is it obvious that Neural Pfaffian is better than the sum of parameterized orbital determinants $\sum_k |(\phi_{i,k}(r_j;r_{-j}))_{i,j}|$ that has a similar number of parameters? By ‘better’, I mean both expressive power (i.e. the least energy that the model can attain with sufficiently long optimization time) and convergence (i.e. the energy achieved within a fixed time period/number of iterations).
2.	Regarding the ablation study on envelopes, it turns out that the full envelope(green) is less costly per iteration than the red and yellow curves. This is weird because the model with efficient envelope contains fewer computations. Furthermore, the red line reaches lower energy than the green line. This is also weird because the model/wave function with full envelope is richer in expressiveness and thus has the potential to attain a lower energy.
Please correct me if I misunderstood this part.
3.	Could the authors give a possible explanation on why Neural Pfaffian’s result gets significantly worse in the 2-hydrogen case? Intuitively this setting is easier than the case with a larger number of hydrogens.
4.	Regarding L157, when the paper addresses systems with odd numbers of electrons, the approach of concatenating an additional learnable vector appears unnatural. The structure of the wave functions would change significantly when a system gains or loses an electron.

Overall, I enjoy reading this work and vote for a strong accept.

**Limitations:**

Yes.

---

> ### Author Rebuttal · Authors · 2024-08-06
>
> We are delighted by the reviewer's positive feedback and want to address the remaining concerns. Firstly, we would like to highlight the broad range of new experimental evidence we present in the general comment. The following details how the experiments relate to the reviewer's concerns.
>
> **Time convergence plot**\
> For our new ablation studies with NeurPf (+ Moon/PsiFormer/FermiNet) and Globe (+ Moon), we added convergence plots regarding compute time in Figure 10 of the general response.
>
> One can see that despite the additional computational overhead through the Pfaffian, our NeurPf is approximately as fast as Globe with the same embedding. This comes mainly from the fewer operations, as we do not require Globe's extra message-passing steps from atoms to orbitals.
>
> **Total energy**\
> We agree with the reviewer that the total energy of all elements in the training set is an imperfect measure. For the ablation study, we decided to plot the total energy as this represents the optimization objective and is less noisy than individual molecular energies. Further, all energies are within the same order of magnitude ($-78.5E_h$ to $-114.49E_h$).
>
> Nonetheless, to better communicate the error per molecule, we added Table 1 in the general response to attribute the error on a per-molecule basis. Generally, models that perform well on one of the molecules also perform well on the others. We also would like to highlight the results in Figure 8 of the manuscript, where we break down the error per molecule with error bars indicating the different structures for finer details. Figure 8 shows that our Neural Pfaffian results in more consistent relative energies than TAO.
>
> **Pfaffian vs. determinant**\
> Great question; we will extend the Appendix with the following discussion to clarify this. While it is a simple result to show that a Pfaffian can generalize Slater determinants, i.e., a Pfaffian can represent every Slater determinant, it is non-obvious that Pfaffians are naturally better in convergence or expressiveness. Empirical evidence in classical QMC [1] finds little to no improvements in molecular systems. However, in non-molecular systems, Pfaffians had greatly improved accuracy [2].
>
> In our new ablation study in Figure 3 of the general response, we replace the learnable $A$ in $\text{Pf}(\Phi A\Phi^T)$ with a fixed one and investigate the impact on convergence. Here, we find that the parametrization is an essential factor in the accuracy of our Neural Pfaffians.
>
> In summary, it is unclear whether Pfaffians are generally better suited for modeling molecular systems. However, as we demonstrate in this work, they can achieve identical accuracy and are well-suited for generalized wave functions.
>
> **Envelopes**\
> We appreciate the keen eye of the reviewer; the classical envelopes are indeed faster than our memory-efficient envelopes. While our envelopes and Pfau et al. [3] aim to reduce memory requirements, they require more operations. In particular, the full envelopes require $O(N_bN_dN_nN_e^2)$ operations. In contrast, our memory efficient envelopes require $O(N_bN_dN_nN_\frac{\text{env}}{\text{atom}}N_e^2)$ (for $N_o=N_e$) operations where $N_b$ is the batch size, $N_d$ is the number of determinants, $N_n$ the number of nuclei, $N_e$ the number of electrons and $N_\frac{\text{env}}{\text{atom}}$ is the number of envelopes per atom in our memory-efficient envelopes.
> Our envelopes primarily reduce the memory from $O(N_bN_dN_nN_e^2)$ for the full envelopes to $O(N_bN_dN_nN_\frac{\text{env}}{\text{atom}}N_e)$ where $N_\frac{\text{env}}{\text{atom}}\ll N_e$.
>
> We most likely attribute the empirical performance to the increased number of wave function parameters. While the $\sigma$ tensor is reduced from $N_d \times N_n \times N_e$ to $N_d \times N_n \times N_\frac{\text{env}}{\text{atom}}$, the $\pi$ tensor is enlarged from $N_d \times N_n \times N_e$ to $N_d \times N_n \times N_\frac{\text{env}}{\text{atom}} \times N_e$. For instance, for $N_e=20, N_n=4, N_d=16, N_\frac{\text{env}}{\text{atom}}=8$, we get the following parameter counts:
> ||$\sigma$|$\pi$|Total|
> |-|-|-|-|
> |full|1600|1600|3200|
> |our|640|12800|13440|
>
> We will update Appendix A to better reflect memory and compute requirements in the context of the full envelopes.
>
> **Hydrogen chain results**\
> We agree with the reviewer that the H2 case is the simplest of all structures. However, it is arguably the most distinct from the other structures as the chain has no "middle" elements. We hypothesize that due to this, it has the lowest accuracy as no fine-tuning has been performed in this experiment.
>
> **Odd numbers of electrons**\
> We are happy to present an alternative solution to deal with odd numbers of electrons in the general comment and would appreciate the reviewer's opinion on this. In short, instead of appending a learnable vector to the orbital matrix, we pad the orbitals $\Phi$ in both dimensions with an identity block to obtain $\hat{\Phi}=\begin{pmatrix}\Phi&0\\\\0&1\end{pmatrix}$. Additionally, we also pad the antisymmetrizer $A$ to $\hat{A}=\begin{pmatrix}A&1\\\\-1&0\end{pmatrix}$ such that one obtains $\text{Pf}(\hat{\Phi} \hat{A}\hat{\Phi}^T)\propto\det\Phi$ if $\Phi$ is square.
>
> We repeat the experiment with this new formulation on the second-row elements experiment in Figure 6 of the general comment. We find little difference. Going forward, we will adopt the new padding technique as it requires no additional parameters.
>
> **Final remarks**\
> We will make sure to correct any typos in our manuscript. We hope to have adequately addressed the reviewer's concerns and questions. We welcome any additional feedback from the reviewer and eagerly await their response.
>
> [1] Bajdich et al. "Pfaffian pairing and backflow wavefunctions for electronic structure quantum Monte Carlo methods"\
> [2] Kim et al. "Neural-network quantum states for ultra-cold Fermi gases"\
> [3] Pfau et al. "Natural Quantum Monte Carlo Computation of Excited States"

---

> > ### Comment · Reviewer_3Wr9 · 2024-08-09
> >
> > Thanks for your reply. I will keep my score.

---

### Official Review · Reviewer_K3cB · 2024-07-11

**Soundness:** 4
**Presentation:** 4
**Contribution:** 4
**Rating:** 6
**Confidence:** 3

**Summary:**

In this paper, the authors propose using Pfaffians instead of Slater determinants to learn a generalized neural wave functions so that the permutation antisymmetry enforcement can be better addressed. Empirical study shows one single proposed model can generalize to various systems (second-row elements) with chemical accuracy. For the nitrogen potential surface prediction, the proposed model outperforms previous work Globe. The proposed model outperforms CCSD(T) and previous work TAO on the TinyMol dataset which contains hundreds of samples.

**Strengths:**

- By replacing Slater determinants, the proposed method avoids discrete and manual orbital selections, and hence is overparameterized, fully learnable, and applicable to any molecular systems.
- Techniques such as memory-efficient envelopes, pretraining by Hartree-Fock, and generalization, are investigated to improve the efficiency and application of the proposed method.

**Weaknesses:**

I am not capable of discovering any weaknesses beyond the ones listed in the Limitation section, or the limitation of the entire neural wave function domain.

**Questions:**

- In Fig. 3, why don't you train the model as many steps as FermiNet did?
- There are other works on generalized wave functions mentioned in the related-work section, such as PESNet. Why aren't they compared in the experiments?
- If I understand correctly, for the large molecules in Fig. 5, none of the neural wave functions outperforms CCSD(T) baseline?

**Limitations:**

The authors provided a section describing the limitations of the current work.

---

> ### Author Rebuttal · Authors · 2024-08-06
>
> We thank the reviewer for their positive feedback on our manuscript and would like to take the opportunity to address the few concerns that were raised. Firstly, we would like to highlight the broad range of new experimental evidence in the general comment. The following details how the experiments relate to the reviewer's concerns.
>
> **Comparison to Globe**\
> To include more baselines in our empirical evaluation, we trained Globe [1] on both TinyMol datasets and plotted the convergence in Figure 1 of the general response. While Globe is initially close to our NeurPf, it converges to higher, i.e., worse energies.
>
> **Comparison to PESNet**\
> We want to stress that PESNet [2] can only perform generalization across different geometric configurations, while Neural Pfaffians tackle the more complex problem of generalization across arbitrary molecular compounds. Nonetheless, we are happy to add comparisons of PESNet on the N2 energy surface. We also added FermiNet [3] from [4] to cover a broader range of methods. However, it should be noted that Globe (Ethene) and our (Ethene) have been trained on a more challenging augmented dataset, while PESNet is only optimized on the N2 energy surface directly. For FermiNet, each structure is optimized independently.
>
> The results in Figure 8 of the general response demonstrate NeurPf's high accuracy on energy surfaces.
>
> **Comparison to CCSD(T) CBS**\
> In Figure 5 of the manuscript, we replicated the setting from [5] and, thus, only trained for 32k steps. However, neural wave functions typically require between 100k and 200k steps to converge [3,6]. Therefore, we extend the training of our Neural Pfaffian to 128k steps and compute energies for the converged model. The results are displayed in Table 1 of the general response. There, our long-trained NeurPf significantly outperforms CCSD(T) CBS on 3 of the 4 large molecules while being within chemical accuracy ($\leq 1.6mE_h$) on the last one.
>
> **Extended training on second-row elements**\
> As the reviewer suggested, we trained our Neural Pfaffian for 200k steps on the second-row elements in Figure 6 of the general response. These results strongly suggest that a single Pfaffian can learn the ionization potentials with higher accuracy with additional training.
>
> **Final remarks**\
> We hope to have answered the reviewer's questions and look forward to an engaging discussion. We appreciate any further feedback and questions from the reviewer.
>
>
> [1] Gao et al. "Generalizing Neural Wave Functions"\
> [2] Gao et al. "Ab-Initio Potential Energy Surfaces by Pairing GNNs with Neural Wave Functions"\
> [3] Pfau et al. "Ab-Initio Solution of the Many-Electron Schrödinger Equation with Deep Neural Networks"\
> [4] Fu et al. "Variance extrapolation method for neural-network variational Monte Carlo"\
> [5] Scherbela et al. "Towards a transferable fermionic neural wavefunction for molecules"\
> [6] von Glehn et al. "A Self-Attention Ansatz for Ab-initio Quantum Chemistry"

---

### Official Review · Reviewer_7TwG · 2024-07-13

**Soundness:** 4
**Presentation:** 4
**Contribution:** 3
**Rating:** 6
**Confidence:** 5

**Summary:**

This paper proposes NeurPf, a novel approach that replaces the standard determinant structure with a Pfaffian-based structure that allows systems of varying sizes to be represented with a single neural wave function. The key idea of the new ansatz is that given a large enough skew-symmetric matrix $A$, we have $\text{Pf}(BAB^\top) = \det(B) \text{Pf}(A)$ for invertible matrix $B$, allowing anti-symmetry to be broadcasted from the neural orbitals to the final output. This makes it possible to set the number of output orbitals $N_o$ into a fixed value that is irrelevant to the system size $N_e$ (so long as $N_e \leq N_o$). Several details on implementing the NeurPf, including architecture selection, envelops, and computation, are discussed. Further, the authors ran experiments on the second-row atoms and small molecular systems to verify the effect of the proposed method.

**Strengths:**

- I think the major contribution, i.e., using Pfaffian to overcome the size consistency issue for varying systems, is quite novel and compelling. It is not only a direct generalization of the Slater determinant (which means that it can be directly applied to most existing architectures), but it allows systems with varying sizes to be represented with one set of parameters.
- The paper is well-written and organized, making it easy to follow and understand.
- The experiments that the authors consider, while not large enough systems (will be discussed later), indeed demonstrate the potential of the proposed method. I especially like the joint training experiments on all atoms of the second row, which serve as solid evidence that the proposed NeurPf ansatz can be used for multiple systems.

**Weaknesses:**

I think the following weaknesses are important to be addressed for the sake of a strong paper:

- First, the authors emphasize the generalization ability of NeurPf, but it looks like all experiments they conduct are a sense of joint training (which I agree is still good to pursue). That said, the authors demonstrate the potential of training one network for multiple models but have not shown that the trained model can be generalized to unseen (but relevant) systems (probably, I admit, with some fine-tuning required). I think it will be necessary for the authors to demonstrate the capacity of generalization since the advantage of NeurPf is exactly to represent multiple systems together. For instance, training on 7/8 of the second row systems and generalizing to the rest, training on the ionized systems of some of the atoms and generalizing to the ionized systems of the rest, etc.
- To me, NeurPf is, instead of a separate algorithm, more of an ansatz modification that can be applied to existing ansatzes, e.g., Ferminet, PsiFormer, LapNet, Moon, and Globe. Applying the method to all those architectures empowers them to be trained jointly on varying systems, as well. Unfortunately, the results of this combination are lacking, and the comparison the authors presented is only between the proposed method applied by Gao et al. 2023a.  I think is will be important if the authors show that
  1. If we apply NeurPf to Ferminet, Psiformer, LapNet, Gao et al. 2023a, and TAO, then the joint training performance is similar to separate training, while the efficiency is much better.
  2. For Fig 5, the results of all existing methods other than TAO should also be presented.
  3. Comparison between NeruPf on different architectures to demonstrate which one is / is not compatible with the proposed architecture.
- Can the authors add a concrete analysis of the computation efficiency of NeurPf? Specifically, how is it compared to a fixed-size Slater determinant (if $N_e = N_o$)? In order to train all systems together, we have to use the largest size (which implicitly increases the computation cost of the smaller systems). How will this influence the overall efficiency? Some plots or tables (instead of a number in texts) are preferred.
- The authors should apply NeurPf on metal atoms (whose structures are complex but similar, and whose ionized energies are important to compute) to see how well it goes. For instance, Training on Na, Mg, Al, K and Ca. These systems are within 20 electrons (smaller than C(NH)_2), and should be feasible to train. The intuition is that since metals share similar electron organization structures to some extent, the proposed method should be able to capture the similarity and hence outperform separate training.

**Questions:**

This is irrelevant to my rating, but right now the number of dimension $N_o$ must be larger than the size of the maximum systems $N_e$. What do you think we should do as the system size $N_e$ goes up to 100, e.g. in Psiformer and LapNet paper?

**Limitations:**

Irrelevant.

---

> ### Author Rebuttal · Authors · 2024-08-06
>
> We thank the reviewer for their invaluable feedback and suggestions. We hope to address their concerns. Firstly, we would like to highlight the broad range of new experimental evidence we present in the general comment. The following details how the experiments relate to the reviewer's comments.
>
> **TinyMol baselines**\
> In addition to the results from [1], we also trained Globe (+ Moon) from [4] on the TinyMol datasets and plot the convergence in Figure 1 of the general response. While starting similarly to our NeurPf, Globe does not converge to the same energies but saturates at energy errors at around $9.4mE_h$ and $47.8mE_h$ on the small and large datasets, respectively.
>
> Unfortunately, we cannot provide FermiNet or PsiFormer energies for all structures as these require $\approx$ 6k A100 GPU hours each.
>
> **Embedding**\
> We agree with the reviewer's suggestion to ablate the embedding method of our NeurPfs. To demonstrate that they also perform well with different embedding methods, we train them with FermiNet [2], PsiFormer [3], and Moon [4] on both TinyMol test sets. We omitted LapNet due to its similarity with PsiFormer but happily add it to the next version of the paper.
>
> The results are depicted in Figure 2 of the general response. NeurPfs outperform Globe and TAO independently of the choice of embedding network. Though consistent with the results from [4], Moon performs best in generalized wave functions.
>
> **Transferability**\
> We agree that transferability is an exciting aspect of generalized neural wave functions but would also like to stress that it has not been the focus of this work as it typically cannot achieve chemical accuracy. Nonetheless, we are happy to present additional experiments. Like Scherbela et al. [1], we first train our NeurPf on the TinyMol training set and then transfer it to the unseen molecules in the test sets.
>
> The results are depicted in Figure 4 of the general response. These suggest that even after pretraining, both methods still require significant fine-tuning to lower errors, but only NeurPf can reach chemical accuracy.
>
> In addition to the TinyMol results, we would like to point the reviewer to the hydrogen chain experiment in Appendix E, where we extrapolate to larger hydrogen chains without finetuning.
>
> **Joint vs. separate training**\
> We compare separately optimized wave functions to our NeurPf trained on the 30 and 40 structures, respectively. Since training separate wave functions for all 70 molecules in TinyMol exceeds our computational resources ($\approx$ 6k A100 hours), we picked one structure for each of the 7 molecules. We trained a separate NeurPf for each to estimate the convergence.
>
> The results are shown in Figure 5 of the general response. At a fixed cost, our generalized wave function generally offers higher accuracy than separately optimizing wave functions. However, we also find that the additional degrees of freedom (higher ratio of parameters/molecule) and specialized optimization offer better final accuracies for separate optimization.
>
> **Computational efficiency**\
> We measure the compute time of the Pfaffian operation in Figure 8 of the general response. Our Pfaffian is five times slower than the determinant. This is primarily due to optimized CUDA kernels for the latter. Note that here, we only measure the Pfaffian and determinant, not the rest of the network.
>
> We benchmark the effect of the batch composition on the time per training step in Figure 9 of the general response for different batches composed of two molecules. There, we see a small overhead for small structures, while for large $N_e$, the time per batch converges to the geometric mean of the individual structures.
>
> When training systems of different sizes, we optimize with various techniques. We work with flattened representations for the embedding network. For the Pfaffian operation (or determinant), we switch to sequential processing for each molecule in a batch (but batch different conformer). This also allows us to use different $N_o$ for each molecule. We want to highlight that this maintains a high level of parallelism thanks to the batch of electronic configurations per molecule.
>
> For a comparison to APG, $\det (\Phi_\uparrow\Phi_\downarrow^T)$, we would like to point the reviewer to our ablation study in Appendix F. AGPs are a special case of Pfaffians. There, the AGP wave function is faster and reaches 32k steps in 70h compared to our Pfaffian's 95h. However, the AGP cannot match the accuracy of our NeurPf.
>
> **Number of orbitals**\
> When going to large systems, the number of orbitals must increase with the number of electrons. As described in Section 4.4 and further detailed in Appendix C.3, we accomplish this by predicting a set of orbitals per atom:
> >[...] we grow the number of orbitals $N_o$ with the system size by defining $N_\text{orb/nuc}$ orbitals per nucleus, as depicted in Fig. 2.
>
> Thus, a system with twice the number of atoms (assuming they are the same atoms) has twice the number of orbitals, while the generalized wave function still has the same number of parameters. The computational scaling of neural network wave functions (incl. FermiNet/PsiFormer/...) to hundreds of electrons remains an issue for future work. Still, NeurPf remains well-defined independent of the system size, thanks to the orbitals growing with system size.
>
> **Final remarks**\
> Again, we thank the reviewer for their detailed assessment and suggestions for improving our manuscript. We hope to have addressed their concerns and look forward to an engaging discussion period. We appreciate any further feedback or questions from the reviewer.
>
>
> [1] Scherbela et al. "Towards a transferable fermionic neural wavefunction for molecules"\
> [2] Pfau et al. "Ab-Initio Solution of the Many-Electron Schrödinger Equation with Deep Neural Networks"\
> [3] von Glehn et al. "A Self-Attention Ansatz for Ab-initio Quantum Chemistry"\
> [4] Gao et al. "Generalizing Neural Wave Functions"

---

> > ### Comment · Reviewer_7TwG · 2024-08-10
> >
> > Thanks for the reply. Part of the concerns have been addressed, but I am not sure why the authors do not reply to the concerns related to training on metal atoms. In terms of the number of electrons, they are approximately ~20 and would not require more resources.

---

> > > ### Author Response · Authors · 2024-08-11
> > >
> > > We are happy to hear that we resolved several concerns. While we intended to accommodate the metal atom experiment in our rebuttal, we found Moon numerically unstable and frequently resulting in NaNs. Given the already large amounts of computation spent on the other rebuttal experiments (>2000 A100 GPU hours), we could not get results for this experiment in time. Nonetheless, we are happy to provide additional experimental evidence on this now that computing resources are available again.
> > >
> > > **Intermediate results on metals**\
> > > As our Neural Pfaffian is independent of the embedding model, we switched to PsiFormer. We then trained on the suggested atoms and their ions. Here, we show intermediate results on the ionization potentials of our Neural Pfaffian compared to the reference energies from [1] after 27k steps (current state of training):
> > >
> > > |          |   Neural Pfaffian (m$E_h$) |   Reference [1] (m$E_h$) |   Error (m$E_h$) | rel. Error   |
> > > |:---------|---------------------------:|---------------------:|-----------------:|:-------------|
> > > | Na |                    189.143 |              188.840 |            0.303 | 0.16%        |
> > > | Mg |                    278.765 |              280.975 |           -2.210 | 0.79%        |
> > > | Al |                    219.486 |              219.958 |           -0.472 | 0.21%        |
> > > | K  |                    160.663 |              159.512 |            1.151 | 0.72%        |
> > > | Ca |                    220.374 |              224.643 |           -4.269 | 1.90%        |
> > >
> > > The Neural Pfaffian energies are averaged over the last 20% of training steps. On 3 of the 5 atoms, the ionization energies are already within chemical accuracy. Note that these are intermediate results, and the model is not yet converged. If the reviewer wishes so, we will update the table as the training continues. In the next iteration of the paper, we will include a similar figure to Figure 3 of the manuscript for the metallic atoms. But, we cannot include it here due to the policies regarding uploading PDFs and external links.
> > >
> > > We attribute Moon's poor performance here to its focus on size-extensivity w.r.t. the number of atoms, which doesn't play a role in atomic systems. Even worse, since Moon heavily relies on message passing between nuclei and electrons, heavier nuclei create information bottlenecks. PsiFormer has no such bottleneck thanks to its self-attention between electrons.
> > >
> > > [1] J.E. Huheey et al. "Inorganic Chemistry : Principles of Structure and Reactivity"

---

> > > > ### Author Response · Authors · 2024-08-13
> > > >
> > > > In light of the upcoming end of the discussion period, we would like to present an update on the ionization energies of the metal atoms. The following table shows the state after 85k steps of training:
> > > >
> > > > |          |   Neural Pfaffian (m$E_h$) |   Reference (m$E_h$) |   Error (m$E_h$) | rel. Error   |
> > > > |:---------|---------------------------:|---------------------:|-----------------:|:-------------|
> > > > | Na (Ion) |                    187.933 |              188.840 |           -0.907 | 0.48%        |
> > > > | Mg (Ion) |                    280.044 |              280.975 |           -0.931 | 0.33%        |
> > > > | Al (Ion) |                    218.970 |              219.958 |           -0.988 | 0.45%        |
> > > > | K (Ion)  |                    157.102 |              159.512 |           -2.410 | 1.51%        |
> > > > | Ca (Ion) |                    222.856 |              224.643 |           -1.787 | 0.80%        |
> > > >
> > > > We would greatly value any feedback or insights the reviewer might have.

---

> > > > > ### Comment · Reviewer_7TwG · 2024-08-13
> > > > >
> > > > > Thanks for the reply. I will raise my score as most of my concerns are addressed.

---

### Author Rebuttal · Authors · 2024-08-06

We thank all reviewers for their invaluable feedback and great suggestions for additional experimental evaluation. We enriched our work with several ablation studies, which we present in the attached PDF. We will add all results to the manuscript.

**Fig 1: TinyMol baselines**\
In addition to the TAO results, we trained Globe [1] on the TinyMol datasets and added it to our evaluation. While Globe is initially close to our NeurPf, it convergences slower and to significantly higher, i.e., worse, energies.

**Fig 2: Embedding**\
Since NeurPf is not limited to Moon, we performed additional ablations with FermiNet [2] and PsiFormer [3] as the embedding.

Our Neural Pfaffians outperform Globe and TAO with any of the three equivariant embedding models. Consistent with [1], Moon is the best choice for generalized wave functions.

**Fig 3: Skew-symmetric construction**\
We picked $\text{Pf}(\Phi A \Phi^T)$ as parametrization because it generalizes Slater determinants and many alternative parametrizations. For instance, by choosing $A=\begin{pmatrix}0 & I\\\\ -I&0\end{pmatrix}$ and $\Phi=(\Phi_1 \hspace{.5em} \Phi_2)$, one obtains the parametrization suggested by Reviewer F2V3 $\text{Pf}(\Phi A \Phi^T)=\text{Pf}(\Phi_1\Phi_2^T - \Phi_2\Phi_1^T)$.
We investigate the impact of having $A$ being fixed/learnable in Figure 3.

The results suggest that having $A$ being learnable is a significant factor in our Neural Pfaffian's accuracy.

**Fig 4: Transferability**\
We want to stress that we focus on direct optimization in this work as it currently provides the only path toward chemical accuracy in generalized wave functions.
Nonetheless, we replicated the setup of TinyMol and pretrain our NeurPf on the TinyMol training set before finetuning on the test sets.
The results show that any method requires significant finetuning. However, only our Neural Pfaffians can match the reference calculations.

**Fig 5: Joint vs separate training**\
We compare separately optimized wave functions to our Neural Pfaffian trained on the 30 and 40 structures, respectively. We plot the total number of steps on the x-axis and the mean difference to CCSD(T) CBS on the y-axis. Since training separate wave functions for all 70 molecules in TinyMol exceeds our computational resources, we picked one structure for each of the 7 molecules. We trained a separate Neural Pfaffian (with Moon) for these 7 to estimate the errors.

At a fixed cost, our generalized wave function generally offers higher accuracy than separately optimizing wave functions. However, we also find that the additional degrees of freedom (higher ratio of parameters/molecule) and specialized optimization offer better final accuracies for separate optimization.

**Fig. 6: Odd numbers of electrons**\
We propose a new solution to address the reviewers' concerns regarding handling odd numbers of electrons. Starting from the classical Slater determinant where $\Phi$ is square and $\Psi=\det\Phi$:

Let $\Phi\in R^{N\times N}$ be the orbitals for odd $N$ electrons and $A\in R^{N\times N},A=-A^T$.
For $\hat{\Phi}=\begin{pmatrix}\Phi&0\\\\0&1\end{pmatrix},\hat{A}=\begin{pmatrix}A&1\\\\-1&0\end{pmatrix},\text{Pf}(\hat{\Phi}\hat{A}\hat{\Phi}^T)\propto\det\Phi$.

In our Neural Pfaffians, we generalize this to $\Phi\in R^{N\times D},A\in R^{D\times D}, \hat{\Phi}\in R^{N+1\times D+1},\hat{A}\in R^{D+1\times D+1}$.

We train our new approach on the second-row elements and show the training energies in Figure 5. As suggested by Reviewer K3cB, we increased the number of training steps to 200k to match FermiNet. The results suggest little difference between the appending of a learnable vector and the new dimension augmentation. Given the avoidance of additional learnable parameters, we use the new parametrization as default.

**Fig. 7: N2 baselines**\
We added FermiNet results from [5] and PESNet [4] as reference energies.

**Fig 8: Pfaffian runtime**\
We benchmark our implementation for $\text{Pf}(\Phi A\Phi^T)$ (incl. the matrix multiplications) against the standard operation of $\det\Phi$ for 10 to 100 electrons. We implement the Pfaffian in JAX while highly optimized CUDA kernels are available for the determinant. In summary, both share the same complexity of $O(N^3)$, but the Pfaffian is approximately 5 times slower.

**Fig 9: Runtime by batch composition**\
Here, we benchmark the total time per step for a two-molecule batch. We test all combinations of two molecules with $N_e^1,N_e^2\in\{2,4,8,16,32\}$. While we find a small runtime increase when processing small molecules jointly, for larger systems, we see the runtime per step converge to the geometric mean of the individual runtimes.

**Fig 10: Convergence by time**\
For NeurPf with FermiNet, PsiFormer, and Moon in addition to Globe (+ Moon), we show convergence by the number of steps. For any time budget, all variants of NeurPf converge to lower energies than Globe.

**Tab. 1: TinyMol energies**\
We list energy differences to CCSD(T) after training for Globe, TAO, and our NeurPf for 32k steps to match the setup from [6]. However, since NN-wave functions typically require 100k-200k steps to converge [2,3], we add a NeurPf trained for 128k steps.

The results show that among generalized wave functions that are optimized on each of the sets, our Neural Pfaffians achieve the lowest energies in 32k steps. Once further converged, our neural Pfaffians also reach or surpass CCSD(T) CBS on the larger structures.

[1] Gao et al. "Generalizing Neural Wave Functions"\
[2] Pfau et al. "Ab-Initio Solution of the Many-Electron Schrödinger Equation with Deep Neural Networks"\
[3] von Glehn et al. "A Self-Attention Ansatz for Ab-initio Quantum Chemistry"\
[4] Gao et al. "Ab-Initio Potential Energy Surfaces by Pairing GNNs with Neural Wave Functions"\
[5] Fu et al. "Variance extrapolation method for neural-network variational Monte Carlo"\
[6] Scherbela et al. "Towards a transferable fermionic neural wavefunction for molecules"

---

### Decision · Program_Chairs · 2024-09-25

**Decision:**

Accept (oral)

**Comment:**

The paper proposes a novel neural many-electron wavefunction ansatz based on the construct of Pfaffian to guarantee the required anti-symmetry. It seems to be the first working neural ansatz other than Slater-determinant-based neural ansatzes, and has the advantage of a unified representation of wavefunctions of different molecular systems (different number of electrons, atom species and positions, etc.). The paper empirically shows that Neural Pfaffians indeed achieve overall more accurate results than existing methods in the cross-system joint training scenario. A few detailed technical designs (e.g., a proper HF initialization) are also presented. The contribution of the paper is acknowledged by all the reviewers. Reviewers also raised a few concerning points, especially the transferrability to unseen molecules, real-time computational cost, and handling odd numbers of electrons, which have been addressed or turned out acceptable given resource limit and the status of the field. Given the potential of opening a new possibility in this field, I recommend accept for this submission.